# INTO THE RABBIT HULL: FROM TASK-RELEVANT CONCEPTS IN DINO TO MINKOWSKI GEOMETRY

**Thomas Fel**[⋆,a,b]    **Binxu Wang**[⋆,a,b]

**Michael A. Lepori**[d]    **Matthew Kowal**[e]    **Andrew Lee**[b]    **Randall Balestriero**[d]    **Sonia Joseph**[g]

**Ekdeep S. Lubana**[b,h]  **Talia Konkle**[a,c]    **Demba Ba**[a,b]    **Martin Wattenberg**[†,b,f]

[a]Kempner Institute, Harvard University    [b]Harvard University
[c]Dept. of Psychology, Harvard University    [d]Brown University
[e]FAR.AI    [f]Google DeepMind    [g]Meta    [h]Goodfire

 kempnerinstitute.github.io/dinovision

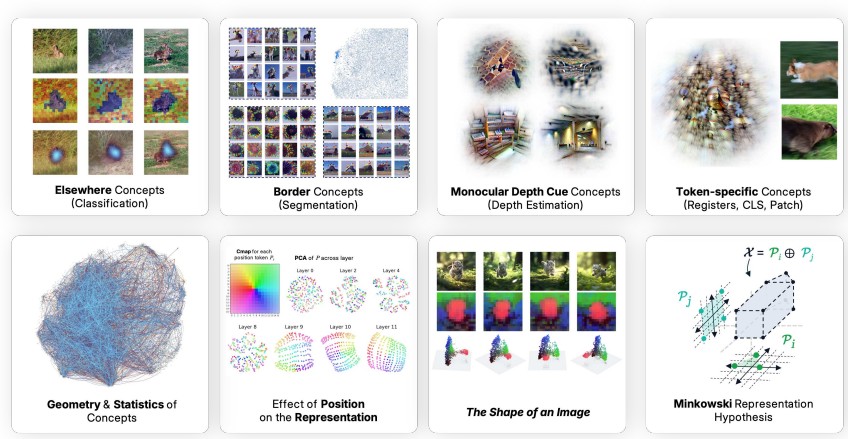

**Elsewhere** Concepts (Classification) — **Border** Concepts (Segmentation) — **Monocular Depth Cue** Concepts (Depth Estimation) — **Token-specific** Concepts (Registers, CLS, Patch)

**Geometry** & **Statistics** of Concepts — Effect of **Position** on the **Representation** — *The Shape of an Image* — **Minkowski** Representation Hypothesis

## ABSTRACT

DINOv2 sees the world well enough to guide robots and segment images, but we still do not know *what* it sees. As a working baseline, we operationalize the Linear Representation Hypothesis (LRH) (features as a sparse combination of near-orthogonal directions) using sparse autoencoders, yielding a 32,000 unit concept dictionary in what constitutes the largest interpretability demonstration for a vision foundation model to date. This method provides the backbone of our study, which unfolds in three parts. First, we analyze how different downstream tasks recruit concepts from our concept dictionary, revealing functional specialization: classification exploits "Elsewhere" concepts that implement "object negation"; segmentation relies exclusively on boundary detectors forming coherent subspaces; depth estimation draws on three distinct monocular cue families aligning with visual neuroscience principles. Turning to concept geometry and statistics, we find higher coherence than random or Grassmannian baselines, sharply decaying spectra with task-aligned anisotropy, antipodal pairs forming signed axes, and low-dimensional token neighborhoods. These patterns reveal partially dense, structured representations that question a purely sparse-coding view of representation. Motivated by these departures, we advance a different view: tokens are formed by combining convex mixtures of a few archetypes (e.g., a rabbit among animals, brown among colors, fluffy among textures). Multi-head attention directly implements this construction, with activations behaving like sums of convex regions. In this picture, concepts are expressed by proximity to landmarks and by regions – not by unbounded linear directions. We call this the *Minkowski Representation Hypothesis*, and present it as a working hypothesis whose testable predictions we outline, together with observed departures from LRH. We conclude by examining how this perspective changes our approach to studying, steering, and interpreting vision-transformer representations.

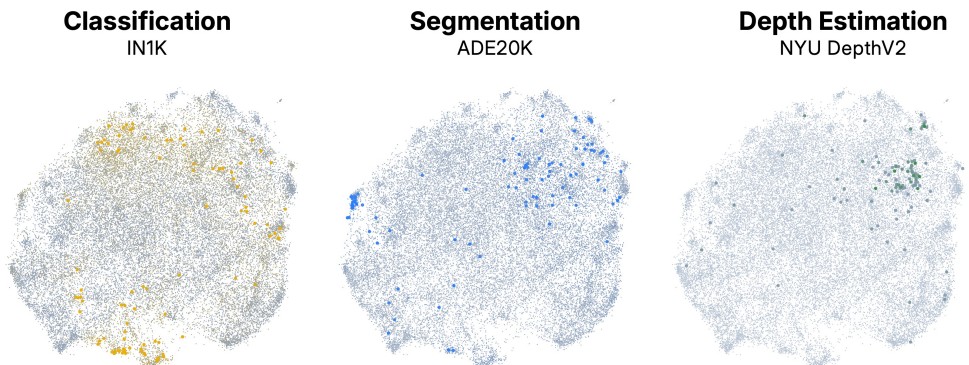

Figure 1: **Concept importance across tasks.** UMAP projection of the learned dictionary, with colors indicating the relative magnitude of each concept's contribution to three downstream tasks: **(Left)** classification (ImageNet-1k), **(Middle)** segmentation, **(Right)** depth estimation. While classification recruits a broad set of concepts, segmentation and depth primarily activate more restricted set of concepts. Although UMAP only preserve local geometry, functionally relevant groupings are visibly clustered in the projection. We show in later sections that different tasks consistently recruit distinct, low-dimensional regions of the concept space.

## 1 INTRODUCTION

Vision Transformers (ViTs) Dosovitskiy et al. (2020) recast images as sequences of patch tokens that are linearly embedded and processed by stacked self-attention and feedforward layers Vaswani et al. (2017), replacing the convolutional inductive bias found in earlier architectures LeCun et al. (2015); Serre (2006). This design scales reliably and flexibly Zhai et al. (2022); Dehghani et al. (2023); Alabdulmohsin et al. (2023), and when trained with contrastive Zhai et al. (2023), masked He et al. (2022), or self-distillation objectives Touvron et al. (2022); Caron et al. (2021), it learns rich, semantically organized representations without labels.

**The case of DINOv2.** DINOv2, trained self-supervised on massive unlabeled data, exhibits strong emergent representations Oquab et al. (2023); Darcet et al. (2023) that support fine-grained classification Chiu et al. (2024), segmentation Liu et al. (2023), monocular depth Mao et al. (2024); Cui et al. (2024), tracking Faber et al. (2024); Tumanyan et al. (2024), and robotic perception Kim et al. (2024). Beyond discrimination, its embeddings serve as priors for generative models Yu et al. (2025), proxies for distributional similarity Stein et al. (2023), and tools to reveal blindspots Bohacek et al. (2025). They are robust and transferable, powering video world models Baldassarre et al. (2025), aligning with language for open-vocabulary segmentation and zero-shot classification Jose et al. (2025), and adapting across domains from satellite imagery Waldmann et al. (2025) to medical scans Ayzenberg et al. (2024); Zhang et al. (2023); Zhou et al. (2024). Yet the internal organization remains unclear: what is encoded, which features are *available* Hermann et al. (2023) to downstream tasks, and how are they geometrically arranged?

**Vision Explainability.** To address these questions, we draw on vision explainability, a field that has developed both an empirical toolkit and theoretical accounts for probing large vision models Doshi-Velez & Kim (2017); Gilpin et al. (2018); Fel (2024). Early work centered on attribution, answering "where" a model looks Simonyan et al. (2013); Zeiler & Fergus (2014); Bach et al. (2015); Springenberg et al. (2014); Smilkov et al. (2017); Sundararajan et al. (2017); Selvaraju et al. (2017); Fong & Vedaldi (2017); Fel et al. (2021); Novello et al. (2022); Muzellec et al. (2024), but these often give surface-level insight and can mislead Hase & Bansal (2020); Hsieh et al. (2021); Nguyen et al. (2021); Colin et al. (2021); Kim et al. (2022); Sixt et al. (2020). To move beyond attribution, concept-based methods have been proposed to extract interpretable latent dimensions Kim et al. (2018); Poeta et al. (2023); Bau et al. (2017); Ghorbani et al. (2019); Zhang et al. (2021); Fel et al. (2023c); Graziani et al. (2023); Vielhaben et al. (2023); Kowal et al. (2024a;b); Fel et al. (2023b). A natural first attempt was to identify individual neurons as carriers of concepts Cammarata et al. (2020). However, neuron-centric accounts struggle with polysemy, basis dependence, and capacity limits Arora et al. (2018); Elhage et al. (2022); Gorton (2024). These limitations motivated methods that identify concepts as distributed features, without requiring alignment to single neurons Ghorbani et al. (2017); Zhang et al. (2021); Fel et al. (2023c). A first theoretical account soon followed in the form of phenomenology by Elhage et al. (2022): in a $d$-dimensional space, neural networks can encode

exponentially many nearly orthogonal features by representing each as a sparse combination of neurons. This phenomenology crystallized into the *Linear Representation Hypothesis (LRH)*, which holds that models contain many more features than neurons, arranged as sparse, quasi-orthogonal directions Park et al. (2024). If the LRH is valid, then concept extraction amounts to an overcomplete dictionary learning problem Fel et al. (2023b). In this framework, the activation space of a layer is factorized into a dictionary basis and sparse codes. Sparse autoencoders (SAEs) Makhzani & Frey (2014); Cunningham et al. (2023); Bricken et al. (2023) are one practical instantiation of this idea, enforcing sparsity through a simple encoder-decoder architecture. When applied to pretrained models such as DINOv2, SAEs uncover a rich library of patterns, which we refer to as *concepts*.

**Our contributions.** We operationalize LRH in DINOv2 with a stable SAE and extract a dictionary of 32,000 concepts, released as (to our knowledge) the largest interactive interpretability demo for a vision foundation model (which will be publicly released upon acceptance). On this basis, our study proceeds in three parts:

- **Downstream usage.** We quantify how tasks recruit the dictionary and find clear specialization: *(i)* classification repeatedly uses "Elsewhere" concepts that implement learned negation; *(ii)* segmentation concentrates on border detectors forming coherent subspaces; *(iii)* depth estimation draws on three families of monocular cues (projective, shadow-based, frequency transitions). These task-aligned subsets are low-dimensional and only weakly overlap.
- **Concept geometry and statistics.** Several diagnostics are compatible with a linear sparse-coding view: atoms are distributed rather than neuron-aligned (low Hoyer scores), and we observe antipodal pairs forming signed semantic axes (so $\cos\theta \approx -1$ often indicates opposite poles of one feature, not unrelated ones). At the same time, the dictionary departs from a near-orthogonal/Grassmannian picture: pairwise similarities have heavier tails, the singular-value spectrum of $D$ decays sharply (anisotropic capacity), and concepts cluster along task-recruited directions. Moreover, positional features are dense yet low-norm, while per-image token clouds remain smooth and low-dimensional even after removing position. Taken together, these results challenge a purely sparse, near-orthogonal "feature packing" account and point to additional geometric constraints.
- **Towards Minkowski geometry.** Guided by these departures, we propose the *Minkowski Representation Hypothesis (MRH)*: tokens behave as sums of convex regions around archetypal landmarks (e.g., animal/rabbit, color/brown, texture/fluffy), and concepts are expressed by proximity to landmarks rather than unbounded directions. We show that multi-head attention constructively realizes MRH (each head produces a convex set; heads sum to a Minkowski sum), and provide preliminary empirical signals consistent with this picture, along with practical implications for steering and structure-aware probes.

With that said, we begin by recalling the LRH and how it can be operationalized through sparse autoencoders, which provide the dictionary of concepts underpinning our analysis.

## 2 LINEAR REPRESENTATION HYPOTHESIS AND OPERATIONALIZATION

A recurring phenomenology of large models is that their representational capacity vastly exceeds the number of neurons: in a $d$-dimensional space, they encode exponentially many features by representing each as a sparse linear superposition Arora et al. (2018); Elhage et al. (2022). Empirically, such features behave as nearly orthogonal directions Papyan et al. (2020), active only in restricted contexts, while neurons themselves are polysemantic Nguyen et al. (2016). The geometrical structure that minimize interference on $c$ vectors in $d$ dimensions is called a Grassmannian frame Strohmer & Heath Jr (2003), and an activation space that we can describe using such object is said to satisfy the Linear Representation Hypothesis Elhage et al. (2022); Costa et al. (2025) (see Appendix B for more details). This motivates two conclusions: (i) neurons are not the appropriate locus of interpretability, and (ii) one must recover the latent basis along which the model effectively operates.

In that view, the LRH is a useful hypothesis precisely because it can be operationalized. If activations admit such a sparse overcomplete representation, then concept discovery reduces to finding the appropriate overcomplete dictionary. While Sparse Autoencoders (SAEs) Cunningham et al. (2023); Bricken et al. (2023) have emerged as a popular choice to learn such a dictionary, they face a persistent challenge in stability: naïve SAEs produce inconsistent features across runs, undermining interpretability Paulo & Belrose (2025); Papadimitriou et al. (2025). To address this, we adopt a stable SAE Fel et al. (2025), which constrains each dictionary atom to lie in the convex hull of real activations. This guarantees that atoms remain in-distribution and yields reproducible, geometrically faithful dictionaries. Formally, let $(\mathcal{X}, \mathcal{F}, \mathbb{P})$ denote the probability space of natural images, $\mathcal{X} \subset$

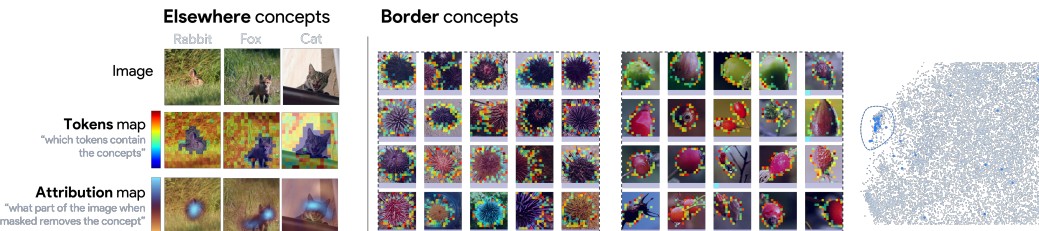

Figure 2: **"Elsewhere" and border concepts.** (Left) In classification, a recurring "Elsewhere" concept fires off-object yet depends on the object's presence, disappearing when the object is removed (via causal masking Petsiuk et al. (2018)), providing evidence suggestive of a causal effect realizing conditional negation (another interpretation being distributed off-object evidence). (Right) In segmentation, top concepts localize along object boundaries, showing consistent spatial patterns and similarity in embedding space, suggesting a shared functional role on a low-dimensional submanifold. See Appendix D for more details and examples.

$\mathbb{R}^{H \times W \times 3}$. For a pretrained Vision Transformer $\boldsymbol{f} : \mathcal{X} \to \mathbb{R}^{t \times d}$, any image $\boldsymbol{x} \sim \mathbb{P}$ yields activations $\boldsymbol{a} = \boldsymbol{f}(\boldsymbol{x}) \in \mathbb{R}^{t \times d}$, i.e. $t$ token embeddings of dimension $d$. For a batch $\boldsymbol{X} = (\boldsymbol{x}_i)_{i=1}^n$, concatenating all tokens gives $\boldsymbol{A} \in \mathbb{R}^{nt \times d}$. Our objective is to factorize $\boldsymbol{A}$ into sparse codes $\boldsymbol{Z} \in \mathbb{R}^{nt \times c}$ and a dictionary $\boldsymbol{D} \in \mathbb{R}^{c \times d}$ with

$$\min_{\boldsymbol{Z}, \boldsymbol{D}} \|\boldsymbol{A} - \boldsymbol{ZD}\|_F^2 \quad \text{subject to} \quad \boldsymbol{Z} \geq 0, \ \|\boldsymbol{Z}_i\|_0 \leq k, \ \boldsymbol{D} \in \text{conv}(\boldsymbol{A}).$$

Here, $\boldsymbol{Z} \geq 0$ denotes an elementwise non-negativity constraint, i.e. all entries of $\boldsymbol{Z}$ are constrained to be nonnegative. Concretely, we used DINOv2-B with 4 registers as $\boldsymbol{f}$, with $d = 768$, $t = 261$. We set $c = 32,000$ atoms and $k = 8$ active codes per token, approximate $\text{conv}(\boldsymbol{A})$ by 128,000 centroids extracted via $k$-means over 1.4M ImageNet-1K images (with augmentation), and parametrize $\boldsymbol{D} = \boldsymbol{SC}$ with $\boldsymbol{S}$ row-stochastic. Codes $\boldsymbol{Z}$ are produced via a single-layer encoder with BatchTopK projection Bussmann et al. (2024); Hindupur et al. (2025). Training with Adam for 50 epochs yields $R^2 > 88\%$ reconstruction fidelity, consistent with prior stability results.

**Interpretation of $\boldsymbol{Z}$ and $\boldsymbol{D}$.** Essentially, the matrix $\boldsymbol{Z}$ encodes the *statistical structure* of the activation space – capturing which concepts are active, how frequently, and to what degree. In contrast, the dictionary $\boldsymbol{D}$ encodes the *geometric structure* defining the atomic directions used to span the space and organize the model's internal feature basis. This decomposition yields a library of 32,000 concept atoms, each interpretable as a linear probe on DINOv2 activations. With this basis in hand, we first investigate which concepts are recruited by different downstream tasks (Section 3), then discuss their statistical and geometric organization in detail (Section 4).

## 3   TASK-SPECIFIC UTILIZATION OF LEARNED CONCEPTS

With the concept dictionary in place, we now ask: *which concepts are actually recruited by downstream tasks?* To answer this, we express linear probes in the concept basis, allowing us to quantify how strongly each concept contributes to a given output. We use this alignment score to compare tasks, while the precise definition and its theoretical justification are deferred to Appendix C.1.

**Different tasks recruit different concepts.** Figure 1 reveals that different tasks recruit different subsets of concepts, often with minimal overlap. Classification activates a wide and dispersed array of concepts, while in contrast, segmentation and depth estimation draw on more compact and localized regions of the concept manifold. This may suggest *functional regions* in the latent space, where concepts are reused non-uniformly across tasks. Quantitatively, we confirm this asymmetry in Figure 11 (Left): classification draws from a broader span of the dictionary than segmentation or depth estimation. In fact, we can show that the recruited concepts seem to bear geometric resemblance. We isolate the top 100 most task-aligned concepts per-head and analyze their pairwise similarities. As shown in Figure 11 (Middle), intra-task concepts are significantly more aligned with one another compared to randomly selected concepts, breaking the quasi-orthogonality observed globally. Finally, in Figure 11 (Right), we confirm this observation by comparing the eigenvalue spectrum of each task's sub-dictionary. All three spectra decay much faster than those of random subsets of concepts, indicating that task-specific concepts form a low-dimensional subspace.

Together, these findings suggest that perceptual tasks selectively activate low-dimensional, functionally specialized subspaces within the broader concept representation space of DINO. But what do

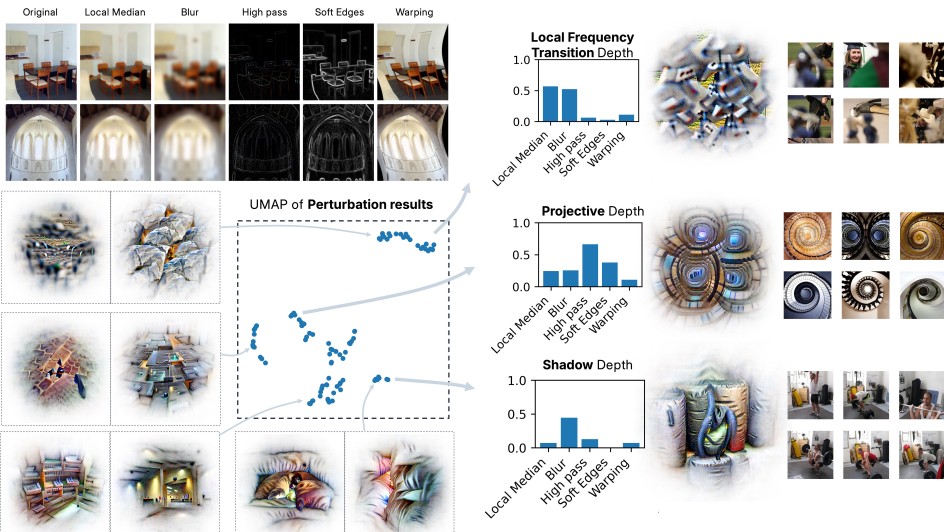

Figure 3: **DINO encodes diverse monocular depth cues.** Visualization of key concepts used in monocular depth estimation tasks. We identify three dominant types: projective geometry cues (e.g., vanishing lines, converging structures), shadow-based cues (e.g., soft lighting gradients and cast shadows), and frequency-based cues (e.g., transitions between high- and low-texture regions). These findings suggest that DINO learns a rich basis of 3D perception primitives from 2D data alone.

these task-specific subspaces actually look like? Can we identify recurring families of concepts that characterize each task? To this end, we now turn to a qualitative examination of these patterns.

**Classification and the *Elsewhere* Concepts.** Across many ImageNet classes, the top concepts for classification include not only objects or parts, but also an "*Elsewhere*" concept (Figure 2, left). These activate broadly across tokens but are suppressed on the object itself, appearing only in surrounding regions. Crucially, they are not generic background detectors: they vanish if the object is removed, indicating a conditional negation—"the object exists elsewhere, but this token is not the object". Such disjoint activations may support classification by outlining boundaries, encoding contrast, or distributing evidence, but they can also mislead attribution maps, which assume concepts are tied to the tokens where they fire. This calls for interpretability tools that combine localization with causal perturbation Shaham et al. (2024). Having explored classification, we now turn to segmentation.

**Segmentation and Border Concepts.** For segmentation, on the ADE20k dataset Zhou et al. (2017), we observe that all the concepts among the top-50 consistently localize along object contours or spatial boundaries. As shown in Figure 2 (right), and expanded in Figure 10, these "border concepts" activate narrowly along the periphery of objects (highlighting limbs, outlines, or silhouette transitions). Remarkably, while the precise visual features vary across classes (e.g., animal ears, tower edges, or sea urchin spines), the spatial footprints of these concepts remain strikingly consistent. Furthermore, in the concept embedding space, these border concepts form a visibly tight cluster (Figure 10, right), suggesting that DINO allocates a dedicated region of its representational geometry to encoding object boundaries. As quantitatively shown in Figure 11, their absolute cosine similarity is higher than average and their eigenspectrum decays faster than a random subset of concepts, suggesting a low-dimensional structure composed of boundary detectors. Segmentation concepts reveal that DINO dedicates portions of its concept space to encoding local spatial structure. We now examine depth estimation, another spatially grounded task, but one that requires global 3D understanding rather than contour localization.

**Depth and Monocular Cue Concepts.** The depth head is trained on NYU depth Nathan Silberman & Fergus (2012) following the original procedure described in DINO (with depth value binning to 256 classes and a linear normalization, see paper for details). Despite no explicit 3D supervision, DINO shows strong performance on monocular depth tasks Mao et al. (2024); El Banani et al. (2024); Zhan et al. (2024). To understand the internal features supporting this, we apply controlled perturbations that isolate specific monocular cues—e.g., median blurring (removes shadows), edge-preserving smoothing (preserves contours), and high-pass filtering (emphasizes projective geometry). We measure concept activation changes and visualize them via UMAP (Figure 3), revealing three functional clusters: (*i*) projective geometry (e.g., vanishing lines), (*ii*) shadow-based (e.g., soft

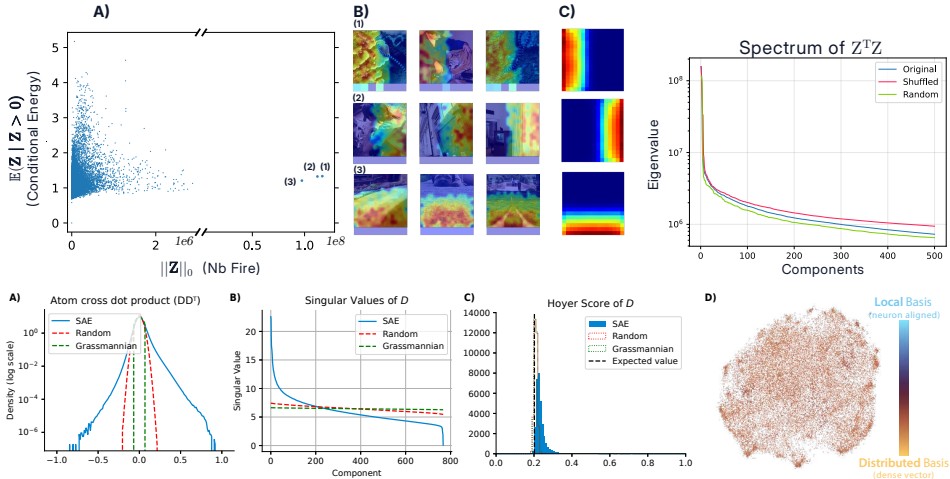

Figure 4: **Concept statistics and geometry. (Top Left)** Conditional energy $\mathbb{E}(\boldsymbol{Z}_i \mid \boldsymbol{Z}_i > 0)$ vs. firing count: most concepts follow a sparse trade-off; a few outliers fire densely with low energy, encoding persistent positional features Sun et al. (2025). **(Top Right)** Spectrum of $\boldsymbol{Z}^\top \boldsymbol{Z}$ shows smooth decay, inconsistent with low-rank modularity but broader than shuffled baselines (Appendix E). **(Bottom)** Dictionary $\boldsymbol{D}$ shows anisotropic structure: (A) heavier-tailed inner products than random/Grassmannian baselines using TAAP algorithm Massion & Massart (2025) (Appendix F); (B) sharp singular value decay; (C) low Hoyer scores confirm distributed (non-neuron-aligned) atoms; (D) UMAP reveals no strong modularity. Overall, DINO's concept space is high-dimensional, distributed, and weakly clustered.

gradients), and (***iii***) local frequency transitions Schubert et al. (2021); Ding et al. (2025), echoing the bokeh concepts of Fel et al. (2024). Some concepts mix cues, as shown in full perturbation maps (Figure 9). These clusters suggest that DINO internally encodes a diverse, interpretable set of monocular depth cues—emerging without labels—and accessible through linear readouts. For more detail see Appendix C.3.

## 4 STASTISTICS AND GEOMETRY OF CONCEPTS

**Concept Occurrence Statistics.** We next examine the statistical profile of concepts. Figure 4 (left) plots conditional energy $\mathbb{E}(\boldsymbol{Z}_i \mid \boldsymbol{Z}_i > 0)$ against firing count. Most concepts follow a triangular envelope: rare ones fire strongly, frequent ones weakly, revealing distinct norm regimes where some features dominate the activation norm while others remain minor. Three outliers break this pattern, showing dense Sun et al. (2025) activations that correspond to positional signals (investigated deeper in Appendix J and Section 5). Thus the space is not strictly sparse: a few universally active, spatially grounded features coexist with a larger ensemble of selective concepts, echoing hybrid sparse–dense regimes Jiang & Lai (2014); Pramanik et al. (2020). **Concepts Co-occurence.** The Gram matrix of concept co-activation $\boldsymbol{G} = \boldsymbol{Z}^\top \boldsymbol{Z} \in \mathbb{R}^{c \times c}$, as shown in Figure 4 (right), has a spectrum that decays smoothly, with no gaps or dominant modes, providing little evidence for modular low-rank structure (concept co-activate without forming clear clusters). Baseline details are provided in Appendix E. **Geometric Organization.** Figure 4 summarizes properties of $\boldsymbol{D}$. Relative to random and Grassmannian baselines, coherence (A) is higher: most atoms are near orthogonal, with small tight clusters, departing from the LRH and suggesting structured redundancy and reuse. The singular value spectrum (B) decays sharply, indicating anisotropic capacity allocation that is consistent with anisotropy in $\boldsymbol{A}$ and local warping from normalization layers Peyré (2009). Hoyer scores (C–D) are far from the one-hot bound, so atoms are distributed rather than neuron aligned Elhage et al. (2022); Colin et al. (2024). Additional motifs such as antipodal pairs are provided in Appendix G. **Concept geometry is only weakly shaped by co-activation.** One hypothesis is that concepts used together should lie nearby in geometry, with co-occurrence "bending" the dictionary into clusters. To test this, we compare the co-activation matrix $\boldsymbol{Z}^\top \boldsymbol{Z}$ with the geometric affinity $\boldsymbol{D}\boldsymbol{D}^\top$. As shown in Figure 13 (middle), they correlate only weakly, suggesting that usage influences geometry but is not

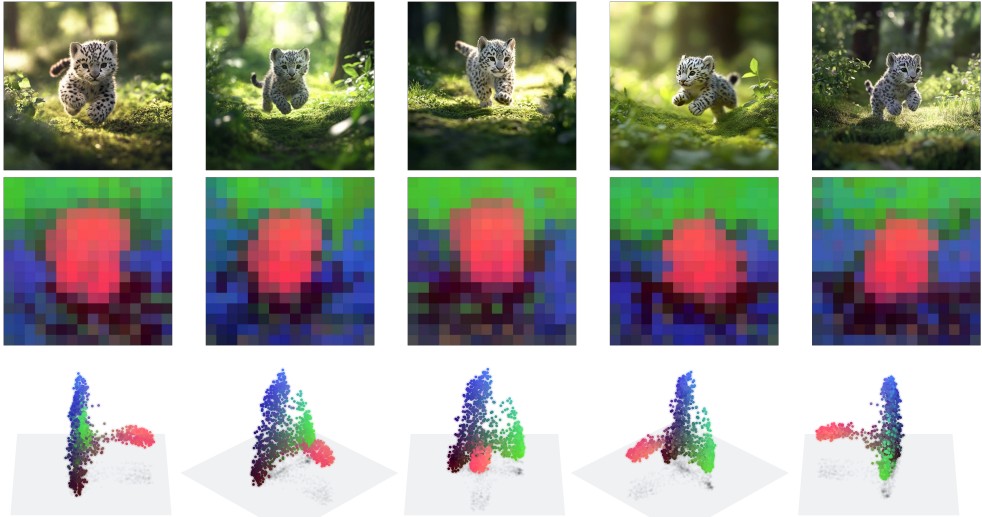

Figure 5: **PCA maps of DINO patch embeddings reveal smooth, semantically aligned structure.** **(Top)** Original images. **(Bottom)** PCA heatmaps of the top three components of patch-token embeddings encoded as $(r, g, b)$ values. Despite no supervision for localization or segmentation, the PCA projections consistently delineate object shapes with smooth transitions across the image grid. Since PCA is a linear operator, it cannot fabricate curvature; the observed smoothness therefore reflects genuine structure in the embeddings. Moreover, the alignment of token geometry across distinct similar images suggest that the representation is not purely relative (i.e., based only on distances within each image tokens).

the dominant organizing principle.[1] Additional evidence comes from UMAP embeddings (Figure 13 right), where top co-activations form tangled, non-local graphs rather than modular clusters.

Taken together, the dictionary is neither maximally incoherent nor uniform: high coherence, sharp spectral decay, task-aligned clusters (Sec. 3), and dense positional signals sit uneasily with a purely sparse, near-orthogonal view. These effects are hard to attribute to simple feature packing and point to additional geometric constraints. We therefore step beyond the SAE lens and examine the model-native token geometry within single images.

## 5 THE SHAPE OF AN IMAGE

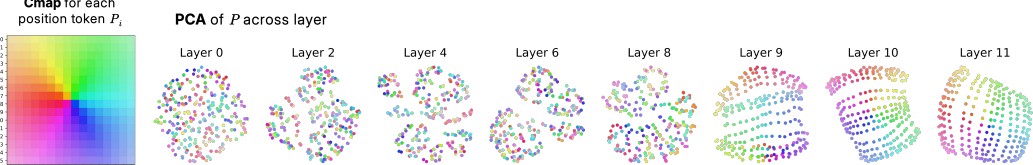

Figure 6: **Visualization of positional encoding across layers reveals smooth compression to a 2D sheet.** PCA projections of the positional encoding vectors at different layers show a clear evolution: from high-rank, dispersed geometry in early layers (Figure 24) to a smooth, low-dimensional sheet in the final layers.

We now study the token geometry within a single image using per-image PCA maps, a standard diagnostic tool that has been employed for feature distillation Oquab et al. (2023); Darcet et al. (2025); Kouzelis et al. (2025), as well as a popular method for showcasing DINO's learned features. In Figure 5, patch embeddings organize along smooth trajectories connecting a small set of extremal points. We begin by testing whether positional information alone explains this structure.

**Just position?** To extract position at each layer, we trained linear decoders to predict token coordinates Islam et al. (2024) and formed a positional basis $\boldsymbol{P} \in \mathbb{R}^{256 \times d}$. Early layers permit near-perfect decoding with high rank, but the positional subspace compresses sharply and approaches

---

[1]Algebraically, the correlation is roughly proportional to the trace of activation covariance $\mathrm{corr}(\boldsymbol{Z}^\top \boldsymbol{Z}, \boldsymbol{D}\boldsymbol{D}^\top) \propto \mathrm{tr}(\boldsymbol{Z}^\top \boldsymbol{Z} \, \boldsymbol{D}\boldsymbol{D}^\top) = \mathrm{tr}(\boldsymbol{D}^\top \boldsymbol{Z}^\top \boldsymbol{Z} \, \boldsymbol{D}) \approx \mathrm{tr}(\boldsymbol{A}^\top \boldsymbol{A}) \propto \mathrm{cov}(\boldsymbol{A})$, which is guaranteed to be positive. This may be an intrinsic property of linear reconstructive methods.

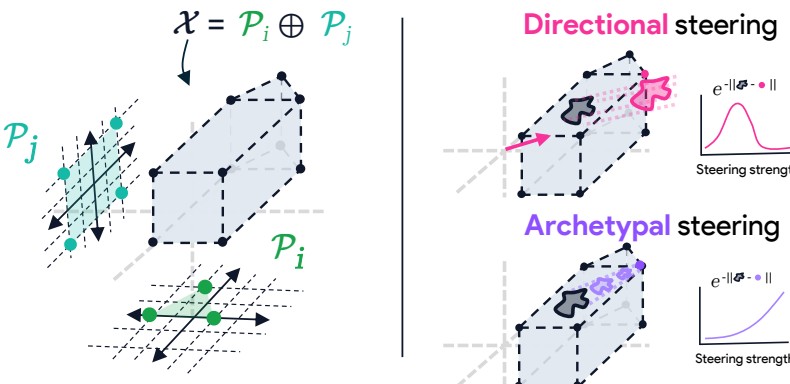

Figure 7: **Minkowski Representation Hypothesis.** Left: the activation set $\mathcal{X}$ is a Minkowski sum of head polytopes $\mathcal{P}_i = \text{conv}(\boldsymbol{A}_{\mathcal{T}_i})$; each head outputs a convex combination of its values and the outputs sum across heads. Right: steering comparison. Directional steering (pink) follows an unbounded vector and leaves the manifold; archetypal steering (purple) moves toward landmarks within polytopes with monotonic proximity.

two dimensions in the final layers (Fig. 24, top), consistent with a shift from place-like to axis coding O'Keefe & Dostrovsky (1971); Chang & Tsao (2017); PCA of $\boldsymbol{P}$ itself confirms this collapse (Figure 6). In real images, positional directions typically appear only among intermediate components (around PCs 3–5; Fig. 24, bottom), so the leading PCs capture non-positional structure. Projecting tokens orthogonally to the positional subspace leaves the PCA organization largely unchanged (Figure 25). This indicates that PCA is capturing something beyond explicit position.

**Toward interpolative geometry.** Since position does not account for the structure we observe, we ask what does. In Figure 5, across many images, patch tokens lie on a consistent low-dimensional set aligned with objects, suggesting interpolation between a few landmark representations rather than purely relative coordinates. DINOv2 offers a concrete mechanism: a DINO head and an iBOT head produce soft assignments over large prototype vocabularies (128k each), yielding mixtures of prototypes for each token Oquab et al. (2023); Zhou et al. (2022). The Kozachenko–Leonenko regularizer further encourages spread in representation space Sablayrolles et al. (2019). A parsimonious view is that tokens are mixtures of a small number of archetypal points drawn from several prototype systems, which yields piecewise smooth variation. In the next section, we propose to formalizes this landmark-based account and shows how attention naturally realizes it.

## 6 Minkowski Representation Hypothesis

We conclude by proposing the Minkowski Representation Hypothesis: a geometry where token embeddings lie in Minkowski sums of convex polytopes spanned by archetypal landmarks. This view is motivated by our observations above and connects to Gärdenfors conceptual spaces in which concepts occupy convex regions Gärdenfors (2004), as well as evidence that concept structure can be convex and compositional Park et al. (2025). Mechanistically, a single attention head outputs a convex combination of its values, and multi-head attention sums these headwise convex sets, yielding a Minkowski sum. Different polytopes can capture factors such as position, depth, or category, with probes reading out proximity to the corresponding landmarks. We now formalize this hypothesis.

**Definition 1.** *Minkowski Representation Hypothesis (MRH). Let $\mathcal{X} \subset \mathbb{R}^d$ be a layer's activation set and $\boldsymbol{A} = (\boldsymbol{a}_1, \ldots, \boldsymbol{a}_c) \in \mathbb{R}^{c \times d}$ an overcomplete archetype set ($c \gg d$) partitioned into tiles $\{\mathcal{T}_i\}_{i=1}^m$, and define $\mathcal{P}_i = \text{conv}(\boldsymbol{A}_{\mathcal{T}_i})$ with $\boldsymbol{A}_{\mathcal{T}_i} = \{\boldsymbol{a}_j : j \in \mathcal{T}_i\}$. MRH holds if (i) $\mathcal{X} = \mathcal{P}_1 \oplus \cdots \oplus \mathcal{P}_m$; (ii) every $\boldsymbol{x} \in \mathcal{X}$ admits a block-convex code $\boldsymbol{x} = \sum_{i \in S} \boldsymbol{A}_{\mathcal{T}_i} \boldsymbol{z}_i$ with $\boldsymbol{z}_i \in \Delta^{|\mathcal{T}_i|}$ and $|S| \ll m$; (iii) for codes $\boldsymbol{Z} \in \mathbb{R}^{n \times c}$ stacked over samples, the Gram $\boldsymbol{G} = \boldsymbol{Z}^\top \boldsymbol{Z}$ exhibits blocks aligned with $\{\mathcal{T}_i\}_{i=1}^m$.*

Put simply, MRH says that a point $\boldsymbol{x}$ is a sparse composition of concept regions, with only a few tiles active $|S| \ll m$, and each active tile contributing a convex combination of its archetypes. For example, a token representing a rabbit might combine: (1) a convex mix from an "animal category" tile (capturing rabbit-like features), (2) a convex mix from a "spatial position" tile (left/center/right),

and (3) a convex mix from a "depth" tile (foreground/background). The final activation is the Minkowski sum ($\oplus$) of these convex contributions.

We now connect the definition to evidence, with full statements and proofs deferred to Appendix K.

**Theoretical account.** First, we show that a single attention head outputs convex combinations of its values, so its attainable set lies in $\text{conv}(V)$ and, under mild reachability conditions, coincides with it (Lemma 1); moreover, affine maps preserve these convex codes (Lemma 2). For multiple head, we naturally find an MRH block-convex structure:

**Proposition 1** (Multi-head attention realizes MRH). *Let there be $H$ heads with value sets $V_h$ and per-head output projections $\boldsymbol{W}_O^{(h)}$. For any input, head $h$ produces weights $\boldsymbol{\alpha}_h \in \Delta^{m_h}$ and output $\boldsymbol{y}_h = \sum_i \alpha_{h,i} \boldsymbol{v}_h^{(i)} \in \text{conv}(V_h)$. After projection and summation,*

$$\boldsymbol{y} = \sum_{h=1}^H \boldsymbol{W}_O^{(h)} \boldsymbol{y}_h = \sum_{h=1}^H \sum_i \alpha_{h,i} \boldsymbol{W}_O^{(h)} \boldsymbol{v}_h^{(i)} \in \oplus_{h=1}^H \boldsymbol{W}_O^{(h)} \big(\text{conv}(V_h)\big).$$

*Thus every output admits an MRH representation with block-convex codes $\boldsymbol{z} = (\boldsymbol{\alpha}_1, \ldots, \boldsymbol{\alpha}_H)$ and archetypes $(\boldsymbol{W}_O^{(1)} V_1, \ldots, \boldsymbol{W}_O^{(H)} V_H)$. If, in addition, each head can realize any point of $\text{relint}(\Delta^{m_h})$ up to the softmax additive constant, then the attainable set is exactly the Minkowski sum.*

Thus, tokens admit block-convex codes over head polytopes and realize MRH with output projections. This mechanism is elementary and aligns with convex partitioning and low-dimensional population geometry in deep networks Montúfar et al. (2014); Raghu et al. (2017); Balestriero et al. (2018); Tvetkova et al. (2025); Tankala et al. (2023); Hindupur et al. (2025); Chung (2021); Cohen et al. (2020), as well as convex conceptual spaces Gärdenfors (2004); Park et al. (2025).

**Empirical evidences.** Next, we test the three MRH criteria on ImageNet-1k tokens. We first compare straight-line interpolation to piecewise-linear $k$-NN geodesics and find that only the latter remain near the data support (Fig. 26, left), consistent with paths that traverse faces of polytopes. Then we compare Archetypal Analysis (AA) Cutler & Breiman (1994) (the $|S| = 1$ case) to an SAE and observe that with about ten archetypes per image AA matches or exceeds SAE reconstruction (Fig. 26, middle), indicating concentration on low-dimensional polytopes Bárány & Füredi (1988); Balestriero et al. (2021). Finally, we examine the Grams of the codes and observe clear block structure (Fig. 26, right), suggesting tiles effect: group of concepts working together under convex constraint K.

**Implications for Interpretability.** If, and this is an assumption, the Minkowski Representation Hypothesis holds, three immediate implications follow. *(i) Concepts are points and regions, not directions.* Under an archetypal view, a concept is a landmark (an extremal point of a convex cell) or a small constellation of landmarks whose hull defines a coherent region. This departs from the linear-factor picture where evidence is the magnitude of an inner product with a preferred direction. *(ii) Steering admits a strict maximum.* In a landmark-based geometry, probing moves an activation toward a specific point or cell; once the landmark (or its convex neighborhood) is reached, the signal saturates and further movement drives the embedding off-manifold. This bounded trajectory helps explain why the gain of SAE-style steering plateaus Wu et al. (2024); Mueller et al. (2025); Karvonen et al. (2025) or reverse Hedström et al. (2025); Templeton et al. (2024) when scaling is pushed too far. Practical probes should therefore estimate proximity to the landmark set (e.g., barycentric weights or geodesic distance within the cell) and stop at convergence rather than extrapolating indefinitely. *(iii) Decomposition is non-identifiable from final activations.* From observations of $\mathcal{X}$ alone, recovering the generating factors (the head polytopes) is ill-posed.

**Proposition 2** (**Non-identifiability of Minkowski decomposition**). *Let $\mathcal{X} = \oplus_{i=1}^m \mathcal{P}_i$ be a Minkowski sum of convex polytopes. Given only samples from $\mathcal{X}$, the decomposition $\{\mathcal{P}_i\}_{i=1}^m$ is generally non-unique: there exist distinct collections $\{\mathcal{Q}_j\}_{j=1}^k$ with $\mathcal{X} = \oplus_{j=1}^k \mathcal{Q}_j$. In particular, even simple polytopes admit infinitely many decompositions as sums of line segments (zonotope generators) with varying directions and lengths.*

See Appendix K.6 and Smilansky (1987) for details. The limitation follows from additivity of support functions under Minkowski addition, $h_{\mathcal{X}}(u) = \sum_i h_{\mathcal{P}_i}(u)$, which admits many valid summands. Practically, this means that estimating individual concept contributions $\boldsymbol{z}$ or polytopes $\mathcal{P}$ from final

activations alone is underdetermined; exploiting intermediate signals (attention weights, per-head outputs) and architectural structure may render the factorization tractable and guides the design of structure-aware interpretability tools.

# 7 DISCUSSION

We trained a RA-SAE on DINOv2 and released a 32k-concept dictionary interactive visualization, showcased in Appendix L. We used this dictionary to quantify how downstream tasks recruit features. We discovered clear specialization for different tasks: "Elsewhere" concepts for classification via learned negation, border concepts for segmentation, and three families of monocular depth cues. Examining concept statistics and geometry, we found the dictionary more coherent than random and Grassmannian baselines, departing from the LRH idealized assumption. We observed dense but low-norm positional concepts suggesting embeddings contain both sparse and dense components. Per-image token clouds showed smooth structure that positional information alone cannot explain. These observations challenged purely sparse, near-orthogonal accounts and led us to propose the *Minkowski Representation Hypothesis*: tokens behave as sums of convex regions around archetypal landmarks; a geometry realizable by multi-head attention through headwise convex combinations. We provide theoretical justification and preliminary empirical evidence, with implications showing that if true, extracting concepts from single layers is insufficient—proper decomposition requires signals from the entire network structure.

While focused on a single architecture, these findings have immediate relevance for the substantial body of research and applications that depend on DINOv2 representations.

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

## A  TABLE OF CONTENTS

# B LINEAR REPRESENTATION HYPOTHESIS

We recall here, briefly, some aspect and motivation behind the Linear Representation Hypothesis (LRH).

Beyond empirical characterization, the geometry suggested by LRH can be motivated by compression: when features are arranged with minimal coherence, it maximizes the number of linearly retrievable features while minimizing destructive interference, a kind of "feature packing".

This principle extends beyond neural activations to fundamental problems in discrete geometry and optimization theory. It connects to classical sphere packing problems: Tammes's problem Mooers (1994), which seeks optimal angular separation of points on a sphere (originally motivated by pollen grain morphology, where surface protrusions must be optimally spaced for aerodynamic dispersal efficiency); Thomson's problem Thomson (1904); Bowick et al. (2002), which minimizes Coulomb electrostatic repulsion energy between charged particles constrained to a sphere; and the spherical code problem Delsarte et al. (1991), which maximizes minimum distance between codewords on the unit sphere for error correction.

These problems share a common mathematical substrate: the geometric structure formalized as Grassmannian frames in signal processing theory Strohmer & Heath Jr (2003). This object underlies the optimization principle of minimizing mutual coherence (the maximum absolute inner product between distinct normalized vectors). These mathematical convergences suggest that neural networks may naturally approximate such optimal geometric configurations to (*i*) maximize representational capacity while (*ii*) minimizing cross-feature interference. This geometric phenomenology crystallizes into our LRH Elhage et al. (2022); Costa et al. (2025) that we formally state here:

**Definition 2.** *Linear Representation Hypothesis (LRH). A representation $\boldsymbol{a} \in \mathbb{R}^d$ satisfies the linear representation hypothesis if there exists a sparsity constant $k$, an overcomplete dictionary $\boldsymbol{D} = (\boldsymbol{d}_1, \ldots, \boldsymbol{d}_c) \in \mathbb{R}^{d \times c}$, and a coefficient vector $\boldsymbol{z} \in \mathbb{R}^c$ such that $\boldsymbol{a} = \boldsymbol{D}\boldsymbol{z}$, under the following conditions:*

$$\begin{cases} \textit{(i) Overcompleteness:} & c \gg d, \\ \textit{(ii) Quasi-orthogonality:} & \mu(\boldsymbol{D}) \equiv \max_{i \neq j} |\boldsymbol{d}_i^\top \boldsymbol{d}_j| \leq \varepsilon, \quad \|\boldsymbol{d}_i\|_2 = 1, \\ \textit{(iii) Sparsity:} & |\operatorname{supp}(\boldsymbol{z})| \leq k \ll c. \end{cases}$$

## B.1 FROM LRH TO DICTIONARY LEARNING.

The LRH induces a natural inverse problem: recover the latent basis on which activations are sparsely expressed. Concretely, given activations $\boldsymbol{A} \in \mathbb{R}^{n \times d}$ (we adopt the row–atom convention $\boldsymbol{D} \in \mathbb{R}^{c \times d}$ used in the main text so that $\boldsymbol{A} \approx \boldsymbol{Z}\boldsymbol{D}$ with $\boldsymbol{Z} \in \mathbb{R}^{n \times c}$), concept extraction becomes a dictionary–learning problem with method-specific constraints on $\boldsymbol{Z}$ and $\boldsymbol{D}$:

$$(\boldsymbol{Z}^\star, \boldsymbol{D}^\star) = \arg\min_{\boldsymbol{Z}, \boldsymbol{D}} ||\boldsymbol{A} - \boldsymbol{Z}\boldsymbol{D}^\top||_F^2,$$

$$\text{s.t.} \begin{cases} \forall i, \boldsymbol{Z}_i \in \{\boldsymbol{e}_1, \ldots, \boldsymbol{e}_k\}, & (\textbf{ACE - K-Means}), \\ \boldsymbol{D}^\top \boldsymbol{D} = \mathbf{I}, & (\textbf{ICE - PCA}), \\ \boldsymbol{Z} \geq 0, \boldsymbol{D} \geq 0, & (\textbf{CRAFT - NMF}), \\ \boldsymbol{Z} = \boldsymbol{\Psi}_\theta(\boldsymbol{A}), ||\boldsymbol{Z}||_0 \leq K, & (\text{SAEs}). \end{cases}$$

Here $\mathbf{I}$ is the identity, $\boldsymbol{e}_j$ denotes a canonical basis vector, and $\boldsymbol{\Psi}_\theta$ is an encoder and a sparsity projection (e.g., TopK, Jump-ReLU, or simply ReLU) producing sparse codes. This formulation unifies the previous clustering-based concept extraction Ghorbani et al. (2019), orthogonal factorization (PCA/ICE) Zhang et al. (2021), nonnegative concept extraction (CRAFT/NMF) Fel et al. (2023c), and modern SAEs Cunningham et al. (2023); Bricken et al. (2023). In practice, these approaches trade off *fidelity* ($\|\boldsymbol{A} - \boldsymbol{Z}\boldsymbol{D}\|_F$) against *sparsity* (e.g., $\|\boldsymbol{Z}\|_0$), yielding a Pareto frontier; SAEs are attractive at scale because the encoder $\boldsymbol{\Psi}_\theta$ enables amortized, batched inference while retaining LRH's sparse, overcomplete structure.

## C   TASK SPECIFIC CONCEPT

In this section, we review additional results and observation on the differents task-specific concepts discussed in Section 3. We will start by giving details on the theoretical root of the importance measure, then we will briefly expand on the "Elsewhere" concept before delving into the monocular depth estimation.

### C.1   IMPORTANCE MEASURE FOR CONCEPT-TASK ALIGNMENT

In the Section 3, we ask which concepts in the dictionary are actually recruited by downstream tasks. We describe here precisely the importance measure we used, which has some appealing property as the linear probe allow us to directly interpret the importance for any linear probe as a linear combination of concepts.

Let $A \in \mathbb{R}^{nt \times d}$ denote token activations (over $n$ images and $t$ tokens per image). We factor $A \approx ZD$ with codes $Z \in \mathbb{R}^{nt \times k}$ and dictionary $D \in \mathbb{R}^{k \times d}$. For a linear probe with weights $W \in \mathbb{R}^{c \times d}$ and predictions $Y = AW^\top \in \mathbb{R}^{nt \times c}$, substituting the factorization gives

$$Y = (ZD)W^\top = Z \underbrace{(DW^\top)}_{W' \in \mathbb{R}^{k \times c}}.$$

The matrix $W'$ encodes the alignment between dictionary concepts (rows) and task outputs (columns). We define the concept-importance vector for the probe as the expected concept activation weighted by this alignment:

$$\phi = \mathbb{E}(Z)\, W' \in \mathbb{R}^c,$$

where the expectation is taken over tokens (and samples) in the evaluation set. Class-wise scores correspond to the components of $\phi$; concept-wise scores can be read from $\mathbb{E}(Z)$ together with the corresponding rows of $W'$.

In linear regimes, this coincides with canonical attribution functionals when expressed in the concept basis. Specifically gradient×input Shrikumar et al. (2017); Simonyan et al. (2013), Integrated Gradients with zero baseline Sundararajan et al. (2017); Ancona et al. (2018), Occlusion Zeiler & Fergus (2014), and RISE Petsiuk et al. (2018) reduce to linear functionals that are proportional to $\mathbb{E}(Z)W'_{:,j}$ when aggregated across tokens. Under standard faithfulness criteria such as C-Deletion, C-Insertion Petsiuk et al. (2018), and C-$\mu$Fidelity Yeh et al. (2019), this is proven to be the optimal attribution when concept are linearly linked to class score; see Theorem 3 in Fel et al. (2023b) (and Ancona et al. (2018) for the initial discussion). We thus use this formulation as a principled and canonical measure of concept importance for linear readouts.

### C.2   ON THE "ELSEWHERE" CONCEPTS

We have observed and discussed in Section 3 a consistent and intriguing pattern: across a wide range of ImageNet classes, the top few most important concepts typically include not only interpretable objects or object-parts, but also an intriguing "*Elsewhere*" concept. As detailed in Figure 8, these concepts activate broadly across the tokens, but crucially *not* on the object itself. Their firing is suppressed exactly where the object appears, and prominent in surrounding regions or background areas. Importantly, "Elsewhere" concepts are *not* generic background detectors: their firing depends critically on the object's presence, and they vanish entirely if the object is removed from the image.

This phenomenon reveals that certain concepts do not fire where the relevant information is located, but instead are able to extract information from one region to fire in spatially distant locations. The Elsewhere concepts represent an extreme example of this spatial decoupling, where the relationship between what drives the concept's firing and where it actually fires exhibits a complete spatial inversion. Rather than simply detecting local features, these concepts implement a form of distributed spatial reasoning that can be characterized as implementing the logical relation "*not the object, but the object exists*". This suggests that DINO has implicitly learned a sophisticated form of fuzzy spatial logic, systematically distributing class-relevant evidence across both object-centric and contextually-related off-object tokens.

The utility of this distributed representation becomes particularly evident when considering the architectural constraints of vision transformers. Since DINO's final classifier operates on the spatial

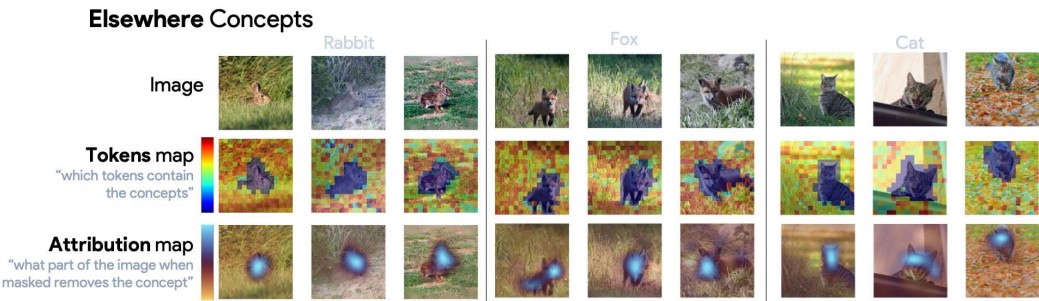

Figure 8: **"Elsewhere" concepts reflect off-object activation conditioned on object presence.** Visualization of a recurring concepts pattern, consistently among the top-3 most important concepts for several ImageNet classes (rows: rabbit, fox, cat), using token-level attribution (middle row) and causal masking Petsiuk et al. (2018) (bottom row). These "Elsewhere" concepts consistently activate in tokens disjoint from the object, yet their presence is conditional on the object itself being present elsewhere in the image: they vanish when the object is removed. Rather than capturing background texture, they express a structured logical relation: "*not the object, but the object exists*". This suggests that DINO implicitly learns a form of fuzzy spatial logic, distributing class evidence across both object-centric and off-object tokens.

average of patch embeddings (concatenated with `cls` token), having class-relevant information distributed across tokens that do not directly contain the object provides several computational advantages. This strategy enhances robustness to partial occlusion, as class evidence remains available in unoccluded regions even when the primary object is hidden. It also provides invariance to viewpoint changes and spatial transformations, since class information is not concentrated solely at object locations. Furthermore, this distributed approach allows the model to integrate multi-scale contextual information that may be crucial for disambiguation in challenging visual scenarios.

To causally verify these interpretations, we employed RISE Petsiuk et al. (2018) analysis, applying random perturbations that mask portions of the image while measuring the resulting changes in concept activation values. The causal attribution maps (bottom row of Figure 8) demonstrate that despite firing in off-object locations, these Elsewhere concepts are causally dependent on the object itself. The RISE attribution is computed as:

$$\Gamma_{\text{RISE}}^{(i)}(\boldsymbol{f}, \boldsymbol{x}) = \mathbb{E}_{\boldsymbol{m} \sim \mathbb{P}_{\boldsymbol{m}}}(\boldsymbol{f}(\boldsymbol{x} \odot \boldsymbol{m}) | \boldsymbol{m}_i = 1)$$

where $\boldsymbol{f}$ represents the composition of the model and the SAE, using 8000 forward passes for each explanation. The results consistently show that the object or animal itself is the most causally important region responsible for concept firing, even though the concept manifests its activation in spatially disjoint locations.

This finding challenges the implicit assumption underlying most heatmap-based concept visualization approaches: that **a concept is primarily *about* the spatial tokens where the concept fires most strongly**. The Elsewhere phenomenon demonstrates a clear dissociation between activation localization and causal attribution, revealing that the most informative regions for understanding a concept's behavior may be spatially distinct from where the concept exhibits its strongest activations. This spatial decoupling would have implications for interpretability research and practice, as it warns against the common tendency to overtrust activation-based visualizations as direct indicators of what information a concept requires or processes.

The prevalence of Elsewhere concepts across diverse object categories indicates that this distributed spatial reasoning is not an artifact of specific classes but represents a fundamental computational strategy employed by vision transformers. This pattern suggests that these models naturally evolve sophisticated spatial logic capabilities that go beyond simple local feature detection, instead developing context-dependent activation patterns that are dynamically modulated by global image content. The discovery of this phenomenon highlights the need for interpretability tools that explicitly account for the potential disconnect between concept activation locations and their causal dependencies, incorporating both spatial activation analysis and causal perturbation methods Shaham et al. (2024) to provide accurate and complete characterizations of learned representations.

### C.3 MONOCULAR DEPTH ESTIMATION

Still in Section 3, We have seen that despite being trained without explicit 3D supervision, DINO exhibits surprising aptitude for depth-related tasks. We have conducted targeted perturbation analysis of depth-relevant concepts using five controlled image manipulations: local median blurring to suppress shadows, global blur to remove fine details, high-pass filtering to emphasize geometric patterns, soft edge enhancement to retain contours, and geometric warping to distort perspective cues. We measure concept activation profiles across these perturbations and project results onto a UMAP embedding showcased in Figure 3.

This systematic analysis reveals three coherent clusters with distinct sensitivity profiles. The local frequency transition cluster responds to blur and median filtering, capturing spatial detail and texture gradients. The projective geometry cluster shows sensitivity to warping and high-pass filtering, detecting perspective lines and structural convergence. The shadow-based cluster exhibits primary sensitivity to median filtering, responding to lighting gradients and cast shadows. **Many concepts exhibit mixed sensitivity profiles**, suggesting DINO learns composite depth representations integrating multiple visual channels. This taxonomy aligns with classical monocular depth cues from visual neuroscience, demonstrating that interpretable 3D perception primitives emerge through self-supervised learning. Full perturbation profiles are available in Figure 9.

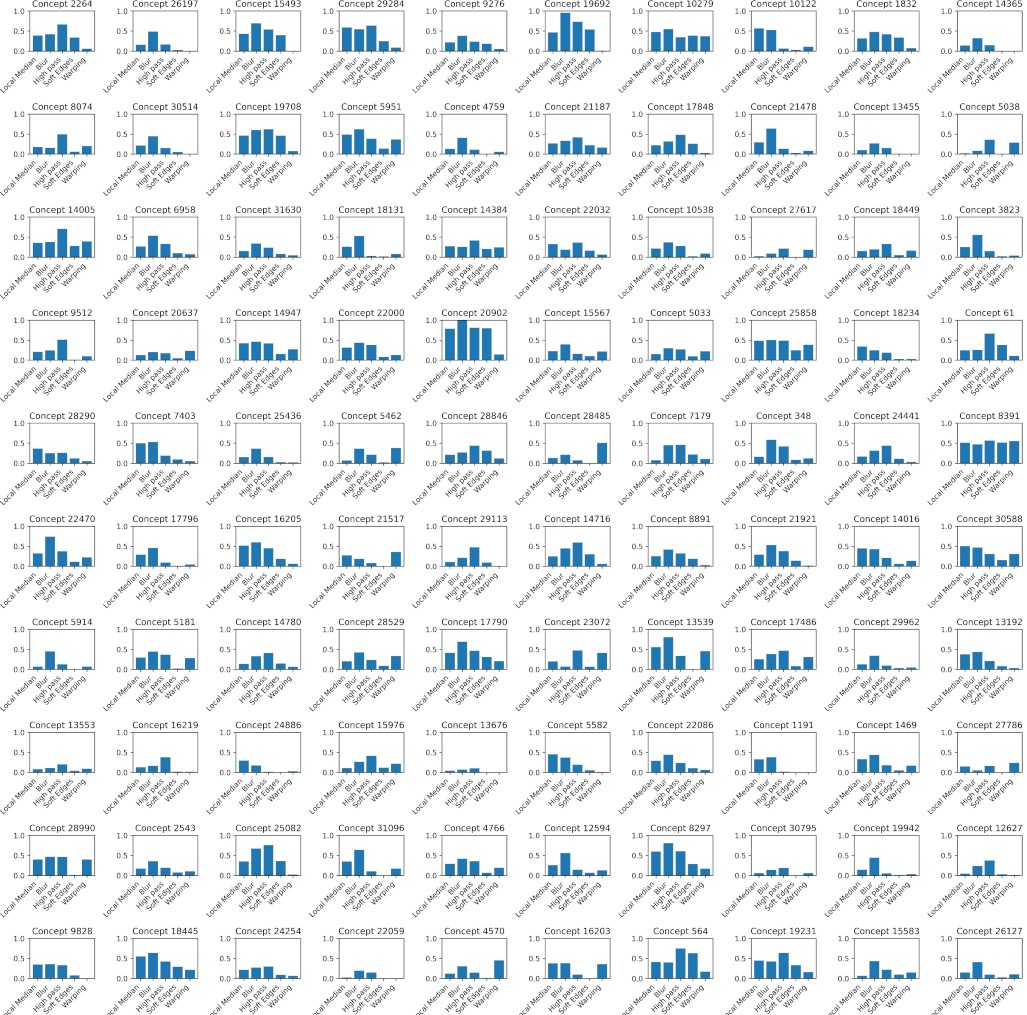

Figure 9: Complete perturbation analysis for depth-relevant concepts showing sensitivity profiles across five image manipulations. The systematic clustering into three main families demonstrates organized depth representation structure within the concept dictionary.

Most Important **Segmentation Concepts**

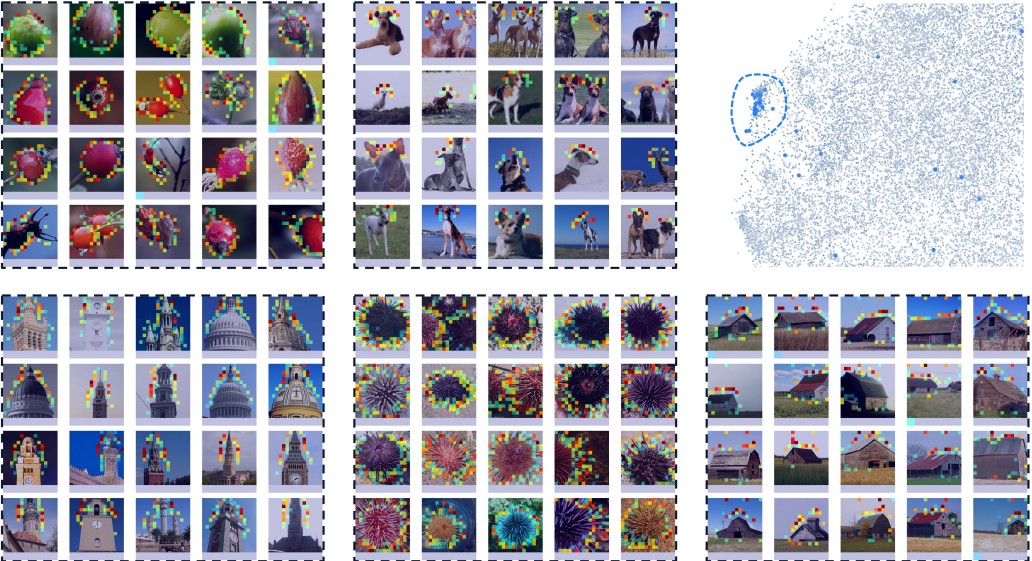

Figure 10: **Segmentation relies on spatially localized border concepts.** Examples of the most important concepts across segmentation tasks, visualized via token attribution (colored overlays). Most of these concepts activate along object boundaries, whether biological (e.g., limbs, heads) or architectural (e.g., domes, rooflines). Despite differences in content, these border concepts exhibit consistent spatial patterns and nontrivial similarity in embedding space (right), suggesting a shared functional role and a possibly low-dimensional submanifold within the concept geometry.

## D TASKS CONCEPT FORM LOW-DIMENSIONAL SUBSPACES

To better understand how different tasks recruit concepts from our learned dictionary, we examine the geometric organization of task-specific concept subsets. While Section 3 demonstrated that different tasks draw from distinct regions of the concept space, here we investigate whether these functional specializations exhibit coherent geometric structure.

We extracted the top-500 concepts most aligned with each task (classification, segmentation, depth estimation) based on their importance scores, as well as a random subset of 500 concepts as a control. For each subset, we computed 2D PCA projections to visualize their geometric arrangement within the concept space.

The results in Figure 12 reveal differences in geometric organization across tasks. Classification concepts are broadly scattered across the projection space, consistent with the diverse range of features needed for multi-class recognition. In contrast, segmentation concepts trace a coherent low-dimensional arc. Depth estimation concepts exhibit clear bimodal structure, reminiscent of the bi-modal organization observed in Figure 11, potentially reflecting the distinct families of monocular depth cues we identified (projective geometry, shadows, and frequency transitions). The random concept subset shows no discernible structure, confirming that the observed patterns reflect genuine functional organization rather than artifacts of the projection method.

These geometric signatures support the hypothesis that task-specific concept recruitment follows principled patterns: each task draws from geometrically distinct subregions of the concept space, with the local geometry reflecting the underlying computational requirements. This functional-geometric correspondence suggests that the concept dictionary exhibits hierarchical organization, where related computational primitives cluster together in representational space.

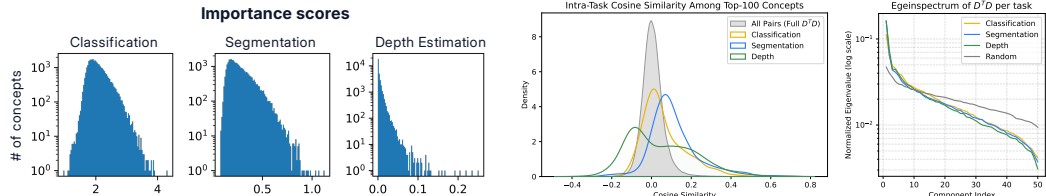

Figure 11: **(Left) Classification recruits more concepts than segmentation than depth.** Classification utilizes a larger fraction of the dictionary compared to segmentation and depth, likely reflecting the higher rank of the classification head. This supports the view that task complexity and output dimensionality shape the breadth of concept recruitment. **(Middle) Intra-task concept similarity.** Cosine similarity histograms of the top 100 most important concepts per task, compared to random subsets of the dictionary. Intra-task concept pairs exhibit higher mutual alignment, deviating from the quasi-orthogonality expected of generic dictionary atoms. This suggests that functional concepts form more coherent subspaces. **(Right) Spectral analysis of task-specific subspaces.** Singular value spectra of the top-100 task-relevant concepts reveal sharply decaying profiles for all tasks (especially segmentation and depth) indicating that each task activates a low-dimensional functional subspace. Compared to random concept subsets, task-aligned subspaces exhibit stronger concentration, supporting a "functional subspace" hypothesis.

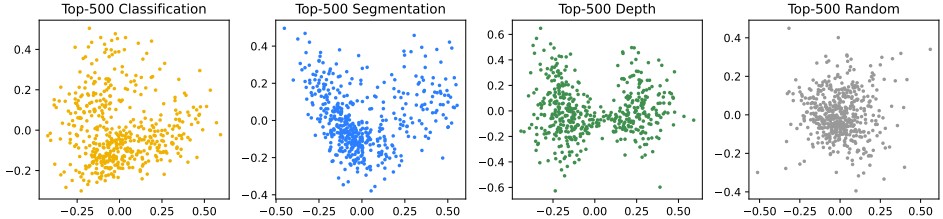

Figure 12: 2D projections (PCA) of top-500 concepts most aligned with each task. Classification concepts (left) are broadly scattered; segmentation concepts (middle-left) trace a low-dimensional arc; depth concepts (middle-right) show clear substructure (bimodality), reminiscent of the bimodal structure in Fig.11 (middle); random concept subset (right) shows no structure. This supports the hypothesis that each task draws from a geometrically distinct subregion of concept space.

## E  BASELINE MODELS FOR CO-ACTIVATION SPECTRUM ANALYSIS

To contextualize the spectral properties of $\boldsymbol{Z}^\top\boldsymbol{Z}$ studied in Section 4, we construct two baseline models that preserve structural characteristics while eliminating semantic organization.

**Random Baseline.** We generate a random symmetric matrix $\boldsymbol{R}$ with identical sparsity $\rho$ and total mass:

$$R_{ij} = U_{ij} \cdot \mathbf{1}(V_{ij} < \rho) \quad \text{where } U_{ij}, V_{ij} \sim \mathcal{U}(0,1) \tag{1}$$

$$\tilde{\boldsymbol{R}} = \frac{(\boldsymbol{R} + \boldsymbol{R}^\top)/2}{\|\boldsymbol{R} + \boldsymbol{R}^\top\|_F} \cdot \|\boldsymbol{Z}^\top\boldsymbol{Z}\|_F \tag{2}$$

with $\mathrm{diag}(\tilde{\boldsymbol{R}}) = \mathrm{diag}(\boldsymbol{Z}^\top\boldsymbol{Z})$. This tests whether the observed spectral structure results from random co-occurrences expected with identical activation sparsity.

**Shuffled Baseline.** We preserve the exact empirical distribution of co-activation strengths while destroying semantic organization through random permutation. Let $\boldsymbol{G} = \boldsymbol{Z}^\top\boldsymbol{Z}$ be the original co-activation matrix. We construct the shuffled baseline as follows:

1. Extract all upper triangular entries: $\mathcal{U} = \{G_{ij} : i < j\}$
2. Apply a random permutation $\pi$ to obtain shuffled values: $\mathcal{U}' = \pi(\mathcal{U})$

3. Construct matrix $S$ where:

$$S_{ij} = \begin{cases} G_{ii} & \text{if } i = j \text{ (preserve diagonal)} \\ u'_k & \text{if } i < j \text{ (where } u'_k \text{ is the } k\text{-th element of } \mathcal{U}') \\ S_{ji} & \text{if } i > j \text{ (copy from upper triangle to preserve symmetry)} \end{cases} \tag{3}$$

This procedure preserves: (i) all diagonal entries (self-activations), (ii) the empirical distribution of off-diagonal values, and (iii) matrix symmetry. However, it destroys the specific concept pairs and potential block structure while maintaining the same marginal statistics as the original co-activation matrix. The shuffled baseline is particularly diagnostic: substantial deviation from $\text{eig}(S)$ indicates that the specific pattern of concept co-activation (not merely the distribution of co-activation magnitudes) carries semantic information.

## F    BASELINE MODELS FOR CONCEPT GEOMETRY ANALYSIS

To contextualize the geometry of dictionary atoms $D$, we also build a few baselines.

**Random vectors on sphere**    We sample a Gaussian random matrix $H \in \mathbb{R}^{c \times d}$, where $H_{ij} \sim \mathcal{N}(0, 1)$ i.i.d., and then we normalized each row to have L2 norm 1. This is reminiscent of the concept vectors when randomly initialized before training. In high dimension, these vectors are usually relatively isotropic on the unit sphere.

**Grassmannian frame**    Next, to stress test the LRH, we numerically computed the Grassmannian from of $c$ atoms in $d$ dimension. This is a non-trivial computational problem, where analytical solution is only available for few scenarios, otherwise we need to rely on iterative optimization to find approximate Grassmannian frames. Brute force gradient optimization of incoherence is slow to converge at our scale. Here, we adapted the algorithm from recent work TAAP Massion & Massart (2025), to further accelerate the solver, we adapted it to CUDA, made Grassmannian frame solving feasible at our problem scale $c = 32000$, $d = 768$. To note, even with GPU acceleration, solving the frame once still takes 6hr on an A100 GPU. In the end, we reached maximal coherence of 0.065897. Indeed, more isotropic and less coherence than the Gaussian random vectors.

# G    ADDITIONAL GEOMETRIC DIAGNOSTICS

In Section 4, we studied the geometry of the dictionary and found that while most dictionary atoms are quasi-orthogonal, we observe deviations that hint at additional non-trivial geometric structure. Some concept pairs are nearly antipodal ($D_i \approx -D_j$), yet correspond to semantically opposed features such as left vs. right or white vs. black. This suggests that DINO may use polarity to encode fine-grained meaning along shared axes (see Figure 13, left).

In addition, still in Section 4 we wonder whether co-activation statistics shape the geometry of the learned concepts. Comparing the concept co-activation matrix $Z^\top Z$ with the geometric affinity matrix $D D^\top$, we find only weak correlation ($r = 0.28$, $R^2 = 0.08$). This indicates that features often used together are not necessarily close in embedding space. A UMAP embedding overlaid with high co-activation links confirms this (Figure 4): connections are non-local, with no strong modularity. Together, these results suggest that DINO's concept geometry is only partially shaped by usage patterns, and that polarity and function may act as latent organizing principles.

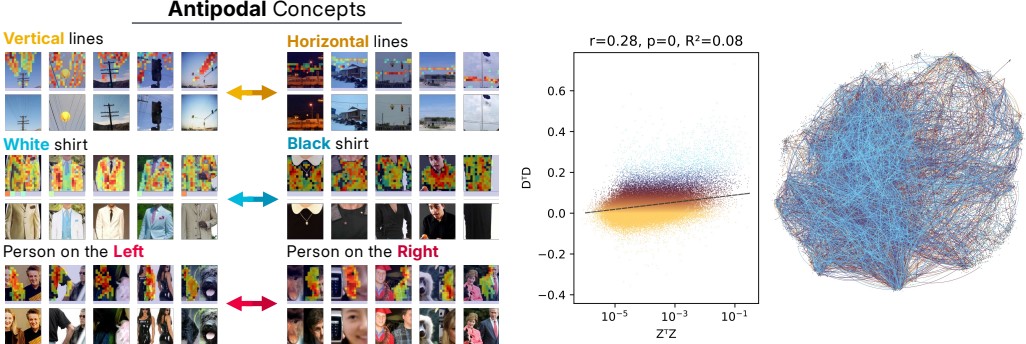

Figure 13: **Emergent geometric patterns in the dictionary. (Left)** Antipodal concept pairs ($D_i \approx -D_j$) encode semantically opposed features (e.g., vertical vs. horizontal). **(Right)** Geometry is weakly shaped by co-activation: correlation between $Z^\top Z$ and $D D^\top$ is low ($r = 0.28$), and UMAP overlays reveal non-local structure with no clear modularity.

# H    TOKEN-TYPE-SPECIFIC CONCEPT

In the first part of this work, specifically the Section 3, we have studied DINO concepts primarily through their semantic content and task alignment. However, this overlooks a fundamental structural aspect in Vision Transformers: the token types. In ViT architectures Dosovitskiy et al. (2020), not all tokens play the same role: `cls` and `reg` tokens are explicitly designed for global processing Darcet et al. (2023), while spatial tokens correspond to local image patches. This raises a natural question: *are some concepts specialized for specific token types? And if so, do they occupy distinct subspaces of the concept geometry?*

## H.1    FOOTPRINT ANALYSIS

To address this, we study the footprint of each concept: the distribution of its activations across token positions. For every concept, we compute the entropy of its token-wise activation over 1.4 million images. Concepts with low footprint entropy are highly localized – activating consistently on specific token subsets – whereas high-entropy concepts are spatially diffuse and positionally agnostic.

Figure 14: **Spatial footprints of 2,500 concepts sorted by entropy**. Each row shows the average activation pattern of one concept across token positions, arranged from most localized (low entropy, top) to most uniform (high entropy, bottom). Localized concepts exhibit strong positional preferences, firing predominantly in specific regions or on particular token types (`cls`, register, or spatial positions), while uniform concepts show broad, distributed activation patterns across the entire image.

Formally, to understand the spatial organization of concept activations, we propose the notion of "footprint" of each concept: its distribution of activations across token positions within images. With $\boldsymbol{Z} \in \mathbb{R}^{nt \times c}$ that we reshape to the tensor $\boldsymbol{Z} \in \mathbb{R}^{n \times t \times c}$ , then for the concept $i$ we compute the footprint $\boldsymbol{\omega} \in \mathbb{R}^t$ as:

$$\boldsymbol{\omega}_i = \frac{1}{N} \sum_{n=1}^{N} \boldsymbol{Z}_{n,:,i}$$

where $\boldsymbol{Z}_{n,:,i}$ is the concept map (a vector of 261 scalars for DINOV2-b) of the concept $i$ on input $n$, and $N$ is the total number of inputs. This captures the average activation strength of each concept at each spatial location, revealing whether concepts exhibit positional preferences or fire uniformly across the image.

We characterize each concept's spatial specificity using the entropy of its empirical footprint distribution. Low entropy indicates highly localized concepts (e.g., firing only at specific positions or token types), while high entropy suggests spatially uniform activation patterns.

The analysis reveals a spectrum of spatial behaviors. Most concepts exhibit relatively uniform firing patterns across spatial tokens, but a significant subset shows strong positional biases. These include concepts that fire exclusively on register tokens (capturing global scene properties like illumination), position-specific concepts that consistently activate in particular spatial regions (potentially encoding geometric or compositional biases), and a single concept that fires exclusively on the `cls` token (likely encoding its positional embedding). This spatial specialization provides further evidence for the functional organization of the concept dictionary, with different concept families optimized for different computational roles within the architecture.

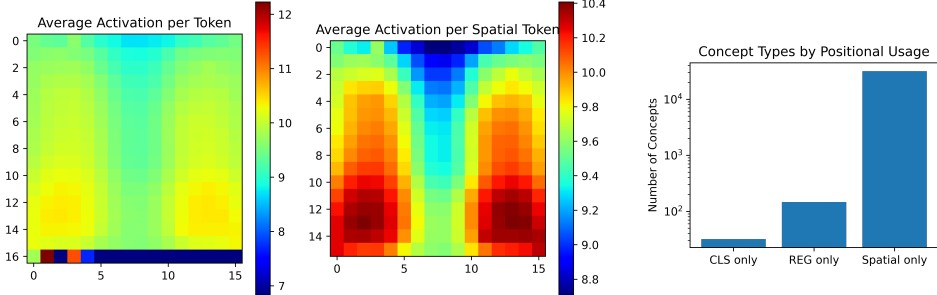

Figure 15: **Statistical analysis of concept footprints**. (Left) Average activation intensity per token position, showing elevated activity on register and `cls` tokens compared to spatial tokens. (Middle) Same analysis restricted to spatial tokens only, revealing subtle positional biases within the image grid. (Right) Distribution of concepts by token-type exclusivity, confirming the findings from Figure 16: one concept fires exclusively on `cls`, hundreds specialize for register tokens, and many are restricted to spatial positions, indicating substantial functional specialization beyond uniform activation patterns.

## H.2 TOKEN-SPECIFIC CONCEPTS.

Figure 16 reveals a continuum of footprint entropy across the concept dictionary. While most concepts are positionally agnostic (high entropy), a distinct subset exhibits highly localized activations. We identify three main categories among low-entropy concepts: (***i***) Position-specific concepts that consistently activate on narrow spatial regions, such as "left-only" or "bottom-only." These may reflect residual positional encoding, biases in the training data, or local geometric primitives. (***ii***) A unique `cls`-only concept, which fires persistently on the `cls` token across all images. This concept is closely tied to the positional embedding of the `cls` token and may act as an "ID" or "passport"-like concepts in the network. (***iii***) A much more diverse (and surprising) group of `reg`-only concepts that activate exclusively on the registers tokens. Unlike the `cls` case, the variety and number of `reg`-specific concepts cannot be explained by position alone. This suggests the register tokens encode a set of non-spatial features that we explore in the next section.

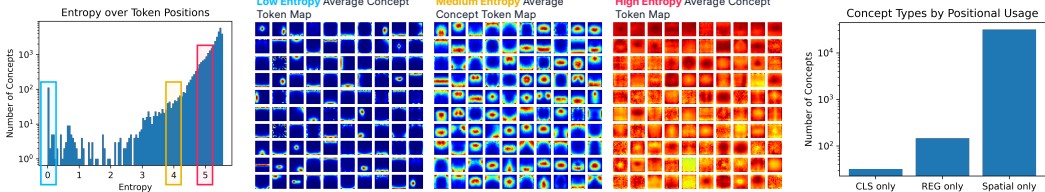

Figure 16: **(Left) Token-wise footprint of concepts.** Distribution of concept entropy across token positions. Most concepts are not token specific (high entropy), but a significant tail exhibits strong localization. The average activation maps for low-entropy concepts often reveal spatial edges (e.g., left/right, top/bottom) or special-token specificity. **(Right) Number of concepts by token-type exclusivity.** While only one concept fires exclusively on the cls token, hundreds are reg-only, indicating substantial specialization beyond positional embedding concept.

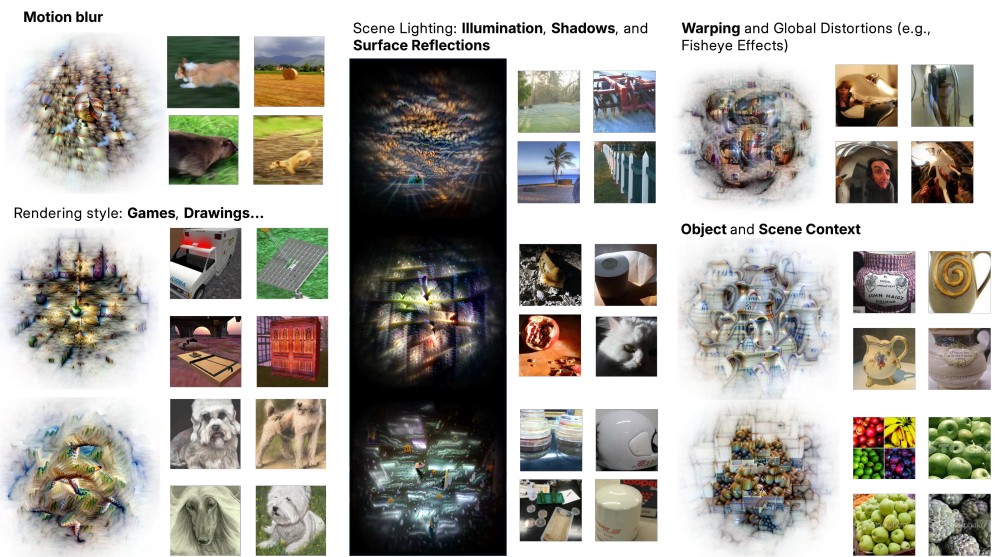

Figure 17: **Register-only concepts capture global, non-local scene properties.** Visualization of selected concepts that activate exclusively on reg tokens. These concepts are not object or region specific, but instead encode global properties such as motion blur, illumination, caustics reflections, lens effects, and style. Their emergence suggests that the register token acts as a potential conduit for abstract scene level information.

**Register-Only Concepts and Global Scene Properties.** Upon inspecting the register-only concepts, an interesting pattern emerges: these concepts do not align with localized object parts or semantic categories. Instead, they seem to encode holistic, global attributes of the image. As shown in Figure 17, register-only concepts respond to phenomena such as lighting style, motion blur, caustic reflections, or artistic distortion. Some even appear sensitive to camera properties (e.g., wide-angle warping or depth-of-field effects).

Interestingly, this specialization is highly asymmetric: while only a single concept activates exclusively on cls, hundreds specialize for reg. This suggests that DINOv2 not only encodes high-level features, but also distributes them across structurally distinct token pathways.

# I   QUALITATIVE VISUALIZATION OF LOCAL GEOMETRY

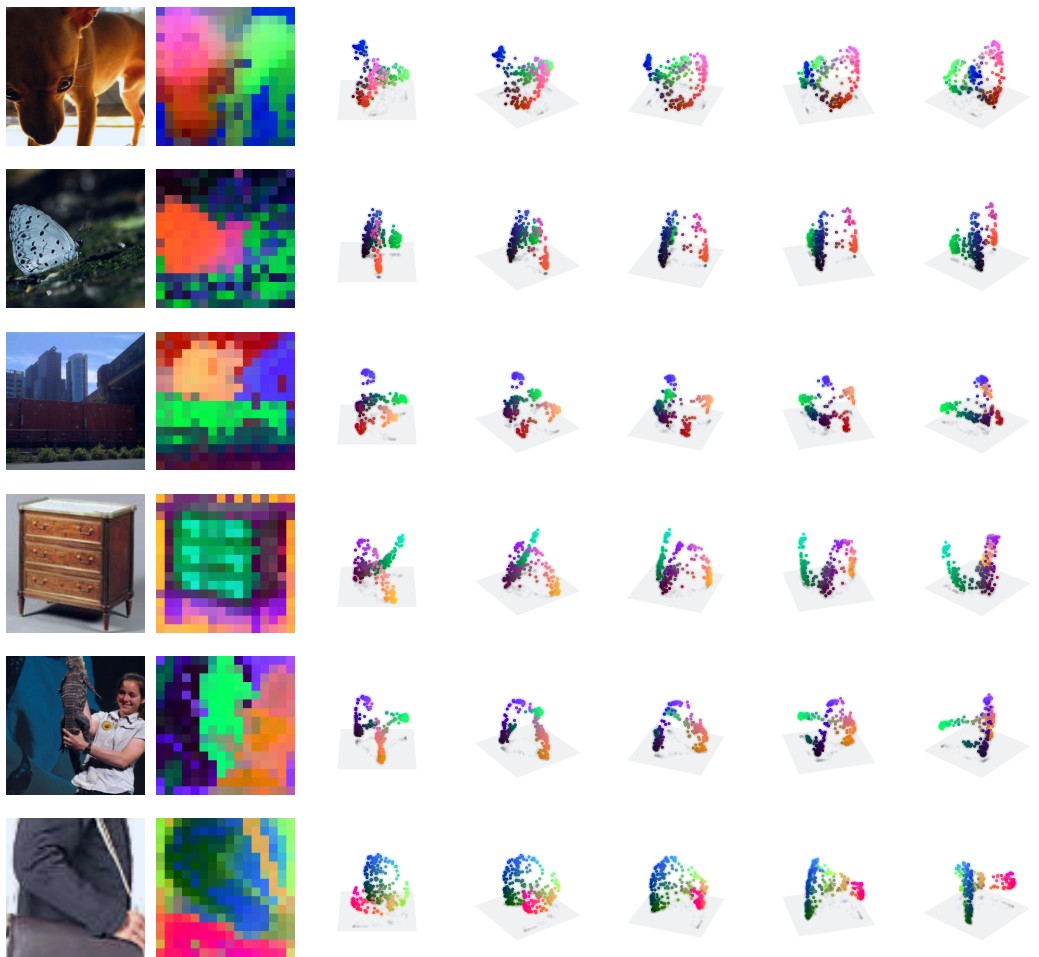

Figure 18: **Local embedding geometry of patch tokens across random ImageNet samples.** Each panel shows a raw ImageNet image alongside its token-level geometry. Patch embeddings from DINO are projected via PCA (on a per-image basis) and visualized by assigning the first three principal components to RGB color (normalized to $[0, 1]$). Despite no supervision, token embeddings seems to lie on smooth, low-dimensional manifolds, with transitions often aligned to object boundaries or perceptual contours.

We provide further visualizations to illustrate the structure of DINO representations at the per-image level. In Figures 18 to 22, patch tokens are projected into their top PCA components, computed independently for each image. The resulting RGB visualizations highlight smooth embedding transitions that often align with object boundaries.

To complement these local views, Figure 19 shows a controlled example across five similar inputs (snow leopards), using a shared PCA basis trained on the entire token corpus (of those 5 images). While individual token positions vary slightly due to pose or lighting, the embeddings align within a common geometric frame. This suggests that DINO not only builds smooth manifolds locally, but does so in a globally consistent latent space.

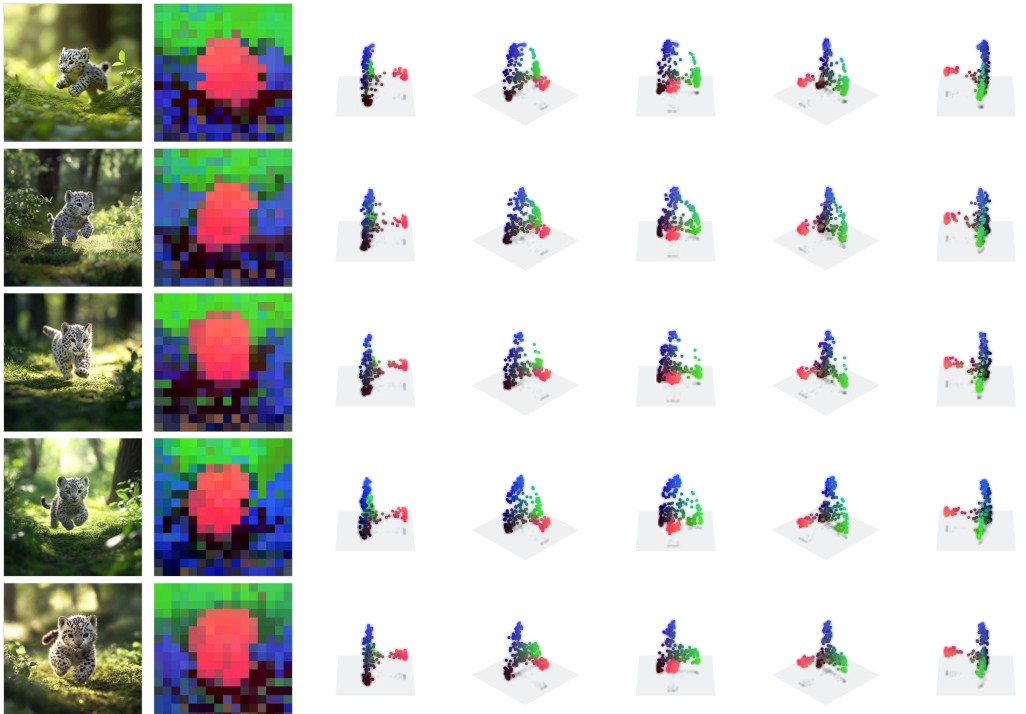

Figure 19: **Detailed structure of token geometry in snow leopard images.** Here we project all patch tokens from five different snow leopard images into the PCA basis trained on all the DINO token pooled across the five images. Each token is visualized as a point in 3D PCA space, and colored in RGB according to its coordinate along the principal components. This shared projection reveals how similarly-positioned objects (here, the leopards) align across images within the same geometric manifold. Despite slight variations in pose and lighting, the representations remain coherent and consistent across instances. This provides evidence for a global manifold structure, within which local image tokenizations trace smooth trajectories.

## J  POSITION EMBEDDING ANALYSIS

To investigate the role of positional information in the observed smooth token geometry, we conduct two experiments: (i) extracting and characterizing the positional basis across layers, and (ii) verifying that smoothness persists even after removing positional information.

### J.1  POSITIONAL BASIS EXTRACTION.

For each layer, we extract positional embeddings from 1 million ImageNet images, yielding $\boldsymbol{A} \in \mathbb{R}^{N \times t \times d}$ token representations where $n$ is the number of images, $t = 261$ tokens per image, and $d = 768$ is the embedding dimension. We employ two approaches to recover the positional basis: **Direct averaging.** We compute the average embedding for each spatial position across all images: $\boldsymbol{p}_i = \frac{1}{N} \sum_{k=1}^{N} \boldsymbol{A}_{n,i}$ where $\boldsymbol{A}_{n,i}$ is the embedding of token at position $i$ in image $n$. We repeat this procedure for each layer. **Linear classification.** We train a linear classifier to predict token position from embeddings, yielding weight vectors $\boldsymbol{w}_i$ for each position at each layer.

Both methods produce highly consistent results: the stable rank profile is similar and the accuracy yield the same results. We therefore choose to use the classifier weights as our primary positional basis for the rest of the experiment.

The analysis in Figure 24 reveals that positional information undergoes systematic compression across layers. Early layers maintain high-rank positional representations that allow precise spatial

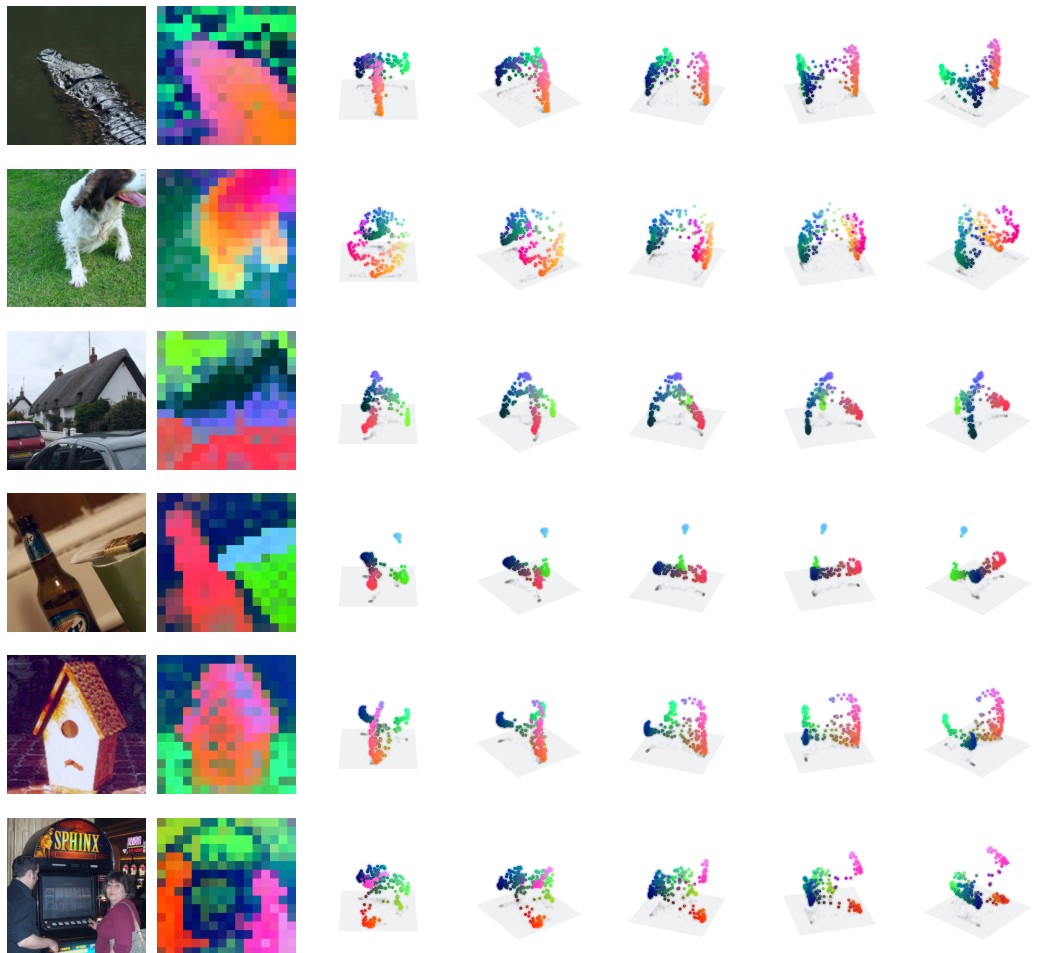

Figure 20: **(cont.)** More examples of PCA-colored patch token embeddings as in Fig. 18.

localization, but this progressively collapses to a low-dimensional (approximately 2D) subspace in the final layers, consistent with a transition from place-cell-like to coordinate-based encoding.

## J.2 STRUCTURE PERSISTS AFTER POSITION REMOVAL

Still in Figure 24, we showed that the position basis is not responsible for the main part of the structure observed in the PCA visualization. To test qualitatively this effect, we project token embeddings orthogonal to the positional subspace, completely removing positional information by projecting the token on the orthogonal subspace of the classifier.

Remarkably, PCA visualizations (in Figure 25) of the original image of Figure 19 embeddings continue to exhibit the same structure, with smooth patterns that align with object boundaries and semantic regions. This demonstrates that the interpolative geometry we observe reflects genuine semantic organization rather than artifacts of positional encoding. The structure emerges from the model's representation of visual content itself, not from spatial coordinate information.

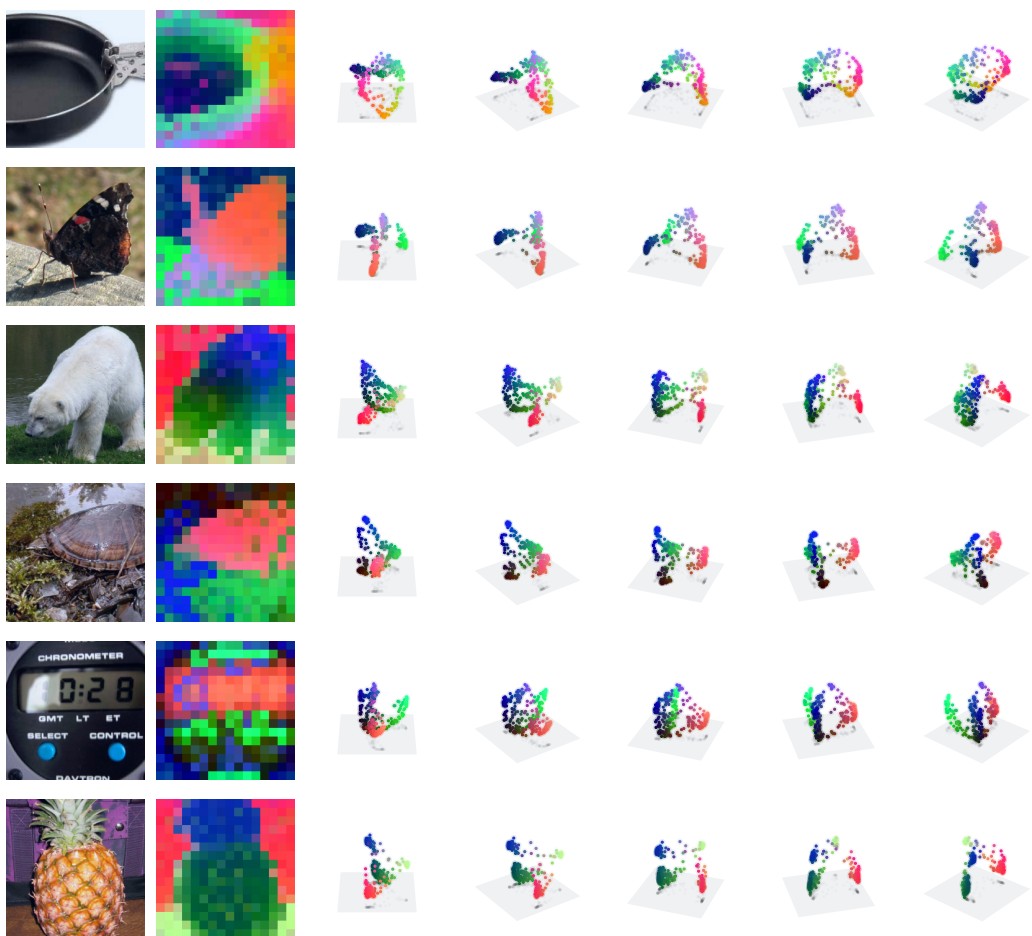

Figure 21: **(cont.)** More examples of PCA-colored patch token embeddings as in Fig. 18.

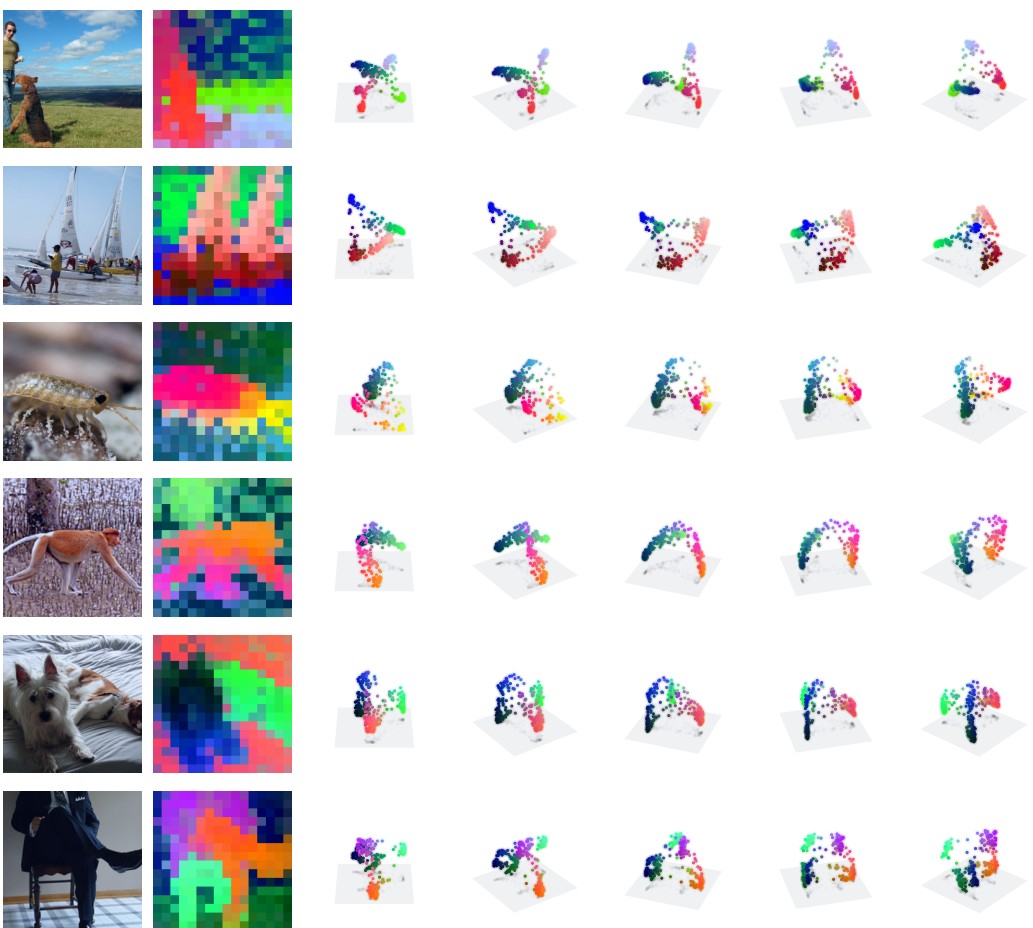

Figure 22: **(cont.)** More examples of PCA-colored patch token embeddings as in Fig. 18.

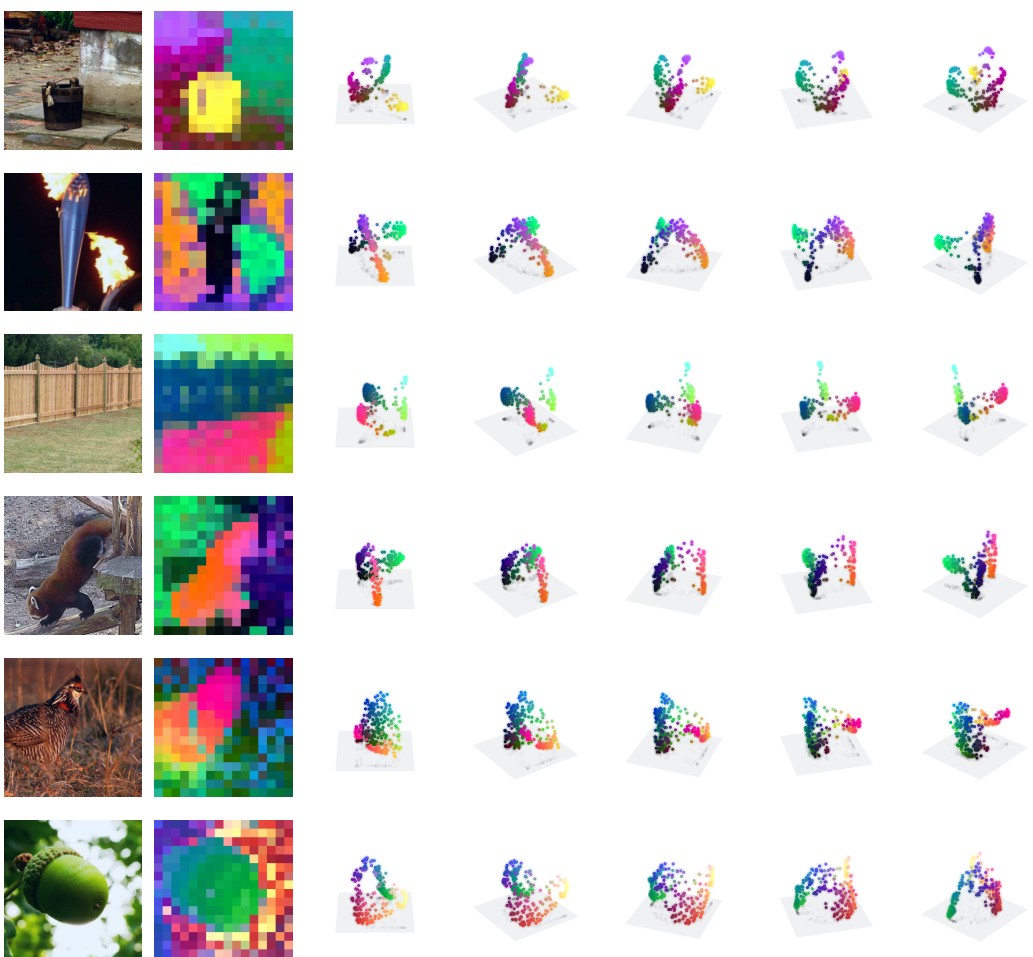

Figure 23: **(cont.)** Final examples of PCA-colored patch token embeddings as in Fig. 18..

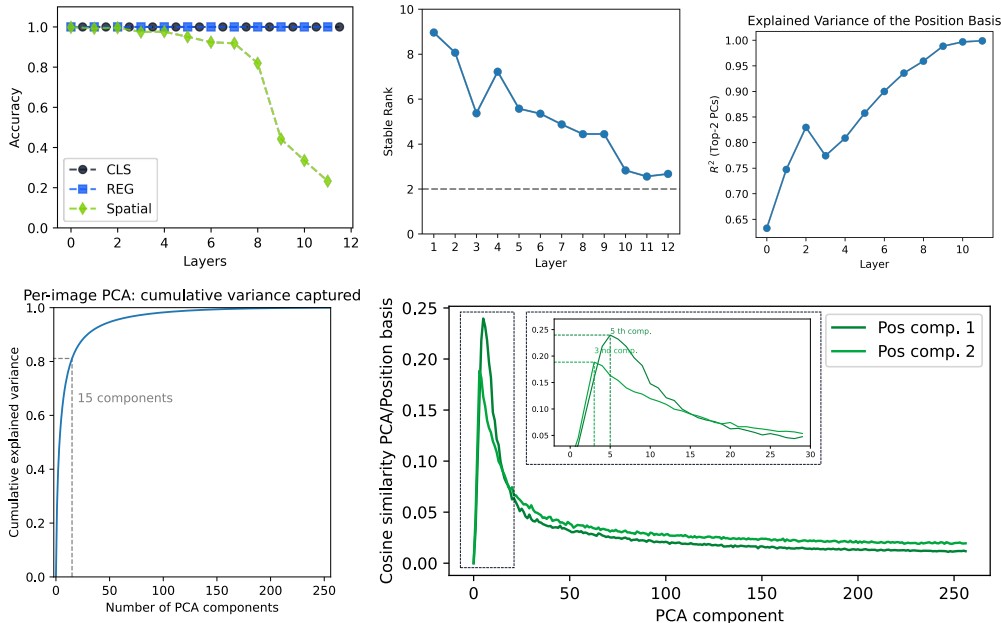

Figure 24: **(Top) Positional encodings compress over layers** (Left) Linear decodability of token position across layers: precise spatial coordinates are recoverable up to layer 8, after which accuracy collapses. `cls` & `reg` tokens are linearly decodable until the end. (Right) The rank of the positional embedding subspace steadily drops, converging to a 2d subspace by the final layers—indicating strong compression. **(Bottom)** Few PCA components of single images tokens explain the variance of tokens across the entire image, suggesting they are lying in a low dimensional subspace. On average, the image-wise PCA shows that position basis correlates with components 3 and 5, but not the dominant directions. The "smoothness" of the PCA maps persists even when positional components are removed, suggesting that this reflects deeper geometric organization beyond position alone.

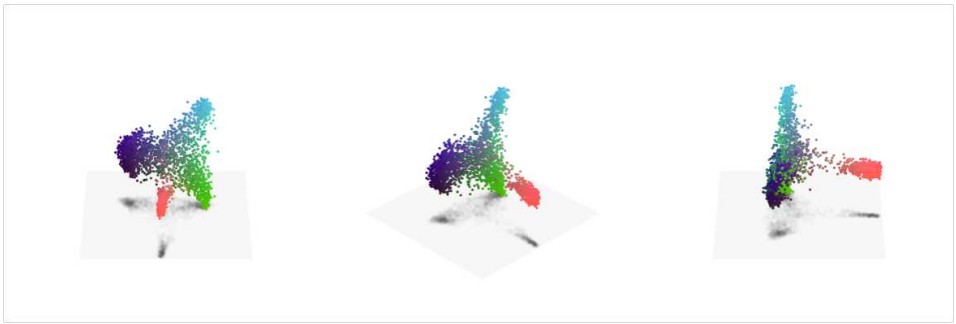

Figure 25: **Smooth token geometry persists after removing positional information**. PCA visualization of patch token embeddings from a rabbit image after projecting orthogonal to the positional basis to completely remove spatial coordinate information. Despite the absence of any positional cues, the token embeddings (colored by their first three PCA components) continue to exhibit smooth, structured patterns that align with object boundaries and semantic regions. This demonstrates that the interpolative geometry reflects genuine semantic organization rather than artifacts of positional encoding.

# K Minkowski Representation Hypothesis

The hypothesis stated in Section 6 is an hypothesis about how the representation space is composed. Such assumption is crucial, as it determines what the method, and what the method can validly recover, (what we can see). To put it simply, if sparse autoencoders implicitly answer to the Linear Representation Hypothesis, then we understand the importance of specifying the right ambient geometry as it not only conditions our interpretation but also determines the very methods we use to extract concepts.

Armed with the observation of Section 4, we contend that an alternative account can explain the phenomena we documented, in particular the interpolative geometry within single images.

## K.1 Background and Related Work on Convex/Polytopal Views

As explained in the main text, our motivation for this hypothesis traces to Gärdenfors' conceptual spaces, where concepts inhabit convex regions along geometric quality dimensions Gärdenfors (2004). Put plainly, we have reasons to believe that the observed interpolation is the surface of a deeper organization: the activation space behaves as a sum of convex hulls [2]. A single attention head already performs convex interpolation over its values, creating an archetypal geometry; multi-head attention then aggregates these convex pieces additively, yielding a Minkowski sum. It is natural to imagine one hull reflecting token position, another depth, another object or part category, so that the final activation is the sum of these convex interpolations, and the "concepts" available to probes are the archetypes governing each hull. We will start by making this idea explicit, then we will review some theoretical evidence based on previous works and showing how simple attention blocks generate such geometry, Then, we will follow showing some empirical signals that make the proposal a plausible candidate and the implication of such geometry. We first formally recall the Minkowski Representation Hypothesis stated in Section 6.

**Definition 3.** *Minkowski Representation Hypothesis (MRH). Let $\mathcal{X} \subset \mathbb{R}^d$ be a layer's activation space and $\boldsymbol{x} \in \mathcal{X}$. Let $\boldsymbol{A} = (\boldsymbol{a}_1, \ldots, \boldsymbol{a}_c) \in \mathbb{R}^{c \times d}$ be an Overcomplete Dictionary of Archetypes $(c \gg d)$. We partition the archetypes into $m$ disjoint tiles $\{\mathcal{T}_i\}_{i=1}^m$, $\mathcal{T}_i \subset \{1, \ldots, c\}$, and define the tile polytopes:*

$$\mathcal{P}_i = \text{conv}(\boldsymbol{A}_{\mathcal{T}_i}) \quad \textit{with} \quad \boldsymbol{A}_{\mathcal{T}_i} = \{\boldsymbol{a}_j : j \in \mathcal{T}_i\}.$$

*Then $\mathcal{X}$ satisfies MRH if*

$$\begin{cases} \textit{(i)} & \textit{Minkowski sum:} & \mathcal{X} = \oplus_{i=1}^m \mathcal{P}_i. \\ \textit{(ii)} & \textit{Block-convex coding:} & \boldsymbol{x} = \sum_{i \in S} z_i \boldsymbol{A}_{\mathcal{T}_i}, \ \boldsymbol{z}_i \in \Delta^{|\mathcal{T}_i|}, \ |S| \ll m. \\ \textit{(iii)} & \textit{Block-structured Gram:} & \boldsymbol{G} = \boldsymbol{Z}^\top \boldsymbol{Z} \in \mathbb{R}^{c \times c} \textit{ with blocks corresponding to } \{\mathcal{T}_i\}_{i=1}^m. \end{cases}$$

*where $\boldsymbol{Z} \in \mathbb{R}^{n \times c}$ stacks archetype codes over $n$ samples and $\Delta^k = \{\boldsymbol{z} \geq 0 : \mathbf{1}^\top \boldsymbol{z} = 1\}$.*

We have formally defined MRH motivated by an empirical observation of interpolative geometry and now ask a basic question: can a single attention block generate such structure? The answer is yes, and the mechanism is elementary. We first review relevant prior work, then show how attention naturally generates such structure. Previous work provides two lines of supporting evidence. First, neural networks naturally partition input spaces into convex regions through their piecewise-linear activations Montúfar et al. (2014); Raghu et al. (2017); Balestriero et al. (2018), suggesting that convex decompositions may be fundamental to deep architectures Tvetkova et al. (2025); Tankala et al. (2023); Hindupur et al. (2025). Analyses have also approached neural networks explicitly through the polytope lens, showing how piecewise-linear partitions structure representation space Black et al. (2022). Second, recent work on neural population geometry shows that deep networks organize representations into low-dimensional manifolds with rich geometric structure Chung (2021); Cohen et al. (2020), which is consistent with a small number of convex factors combining to yield the observed variability. Third, in the language domain, categorical and hierarchical concepts have been shown to admit convex (polytopal) representations, with semantic relations reflected directly in geometric structure Park et al. (2025).

---

[2]This aligns with evidence that concept structure can be convex and compositional in other domains Park et al. (2025)

We now demonstrate that standard attention mechanisms naturally generate the geometry described by Definition 1. The argument proceeds in three steps, and full proofs are provided in Appendix K. showing how elementary operations compose to create the previously describe geometric structure.

We will show that the elementary operations available in DINO, and in particular a standard multi-head attention block, already generate the geometry described by the Minkowski Representation Hypothesis. We then state and prove a few basic properties of this geometry and discuss its robustness. We begin by recalling some basic fact and with elementary results.

**Notations.** For a finite set of vectors $\boldsymbol{V} = \{\boldsymbol{v}_1, \ldots, \boldsymbol{v}_m\} \subset \mathbb{R}^d$ the convex hull $\mathrm{conv}(\cdot)$ is

$$\mathrm{conv}(\boldsymbol{V}) = \Big\{ \sum_{j=1}^m \alpha_j \boldsymbol{v}_j \ : \ \boldsymbol{\alpha} \in \Delta^m \Big\} \quad \text{where} \quad \Delta^m = \Big\{ \boldsymbol{\alpha} \in \mathbb{R}^m \ : \ \alpha_j \geq 0, \ \sum_{j=1}^m \alpha_j = 1 \Big\}.$$

And we denote by $\boldsymbol{\sigma} : \mathbb{R}^d \to \mathbb{R}^d$ the standard softmax:

$$\sigma(\boldsymbol{x})_j = \frac{e^{x_j}}{\sum_{k=1}^m e^{x_k}}$$

and recall two basic properties that will be used repeatedly: (*i*) softmax is invariant under adding a constant along the all-ones direction, $\boldsymbol{\sigma}(\boldsymbol{x}+\lambda\boldsymbol{1}) = \boldsymbol{\sigma}(\boldsymbol{x})$, (*ii*) it maps $\mathbb{R}^m/\mathrm{span}\{\boldsymbol{1}\}$ diffeomorphically onto the relative interior of the simplex $\mathrm{relint}(\Delta^m)$, with inverse given by the log map up to an additive constant, namely if $\boldsymbol{\alpha} \in \mathrm{relint}(\Delta^m)$ then $\boldsymbol{x} = \log \boldsymbol{\alpha} + \lambda\boldsymbol{1}$ satisfies $\boldsymbol{\sigma}(\boldsymbol{x}) = \boldsymbol{\alpha}$.

## K.2 FROM A SINGLE HEAD TO CONVEX POLYTOPES

We start with the most basic question: *what does a single attention head produce geometrically?* A single head takes a query, forms attention weights over a fixed set of value vectors, and returns a weighted sum of those values. We now make precise the fact that the range of this map is a convex polytope, namely the convex hull of the values, and that under a mild reachability condition every point of that hull can be attained.

**Lemma 1** (Single head creates convex polytopes). *Consider one attention head with values $\boldsymbol{V} = \{\boldsymbol{v}_1, \ldots, \boldsymbol{v}_m\} \subset \mathbb{R}^d$ and attention weights $\boldsymbol{\alpha} \in \Delta^m$. The attainable output set is*

$$\mathcal{Y} = \Big\{ \sum_{j=1}^m \alpha_j \boldsymbol{v}_j \ : \ \boldsymbol{\alpha} \in \Delta^m \Big\} \subseteq \mathrm{conv}(\boldsymbol{V}).$$

*Moreover, suppose the pre-softmax logit map can generate any vector in $\mathrm{Im}(\boldsymbol{K}^\top) + \mathrm{span}\{\boldsymbol{1}\} = \mathbb{R}^m$ as the query varies, where $\boldsymbol{K}$ denotes the matrix of keys and the invariance to $\boldsymbol{1}$ reflects softmax's additive invariance. Then $\mathcal{Y} = \mathrm{conv}(\boldsymbol{V})$. In this case they admit a MRH representation, their codes are $\boldsymbol{z} = \boldsymbol{\alpha}$ and the archetypes are $\mathcal{A} = \boldsymbol{V}$.*

*Proof.* By definition of a single head, the output has the form $\boldsymbol{y} = \boldsymbol{\alpha}\boldsymbol{v}$ with $\alpha \in \Delta^m$. Therefore $\boldsymbol{y} \in \mathrm{conv}(\boldsymbol{V})$ and $\mathcal{Y} \subseteq \mathrm{conv}(\boldsymbol{V})$. Now, let $\boldsymbol{\alpha} \in \mathrm{relint}(\Delta^m)$ be arbitrary. Define $\boldsymbol{u} = \log \boldsymbol{\alpha}$ entrywise and note that adding any constant $c$ along $\boldsymbol{1}$ leaves softmax unchanged. Hence $\boldsymbol{\sigma}(\boldsymbol{x} + \lambda\boldsymbol{1}) = \boldsymbol{\alpha}$. If the pre-softmax logit map can realize any vector in $\mathbb{R}^m$ up to the $\boldsymbol{1}$ direction, then there exists a query producing logits $\boldsymbol{x} + \lambda\boldsymbol{1}$. For that query, the attention weights equal the prescribed $\boldsymbol{\alpha}$ and the head output is the convex combination $\sum_j \alpha_j \boldsymbol{v}_j$. The argument above covers all points with strictly positive coefficients, that is $\mathrm{relint}(\Delta^m)$. The extreme points and boundary faces of $\Delta^m$ can be attained by limits of interior points (or by sending some logits to $-\infty$), hence their images under the affine map $\boldsymbol{\alpha} \mapsto \sum_j \alpha_j \boldsymbol{v}_j$ are limits of attainable outputs. Therefore $\mathcal{Y}$ contains all of $\mathrm{conv}(\boldsymbol{V})$.

Finally, for the MRH identification, this corresponds to the special case of $|S| = 1$ with a single polytope, where the codes are simply $\boldsymbol{z} = \boldsymbol{\alpha}$ and the archetypes are the values $\mathcal{A} = \boldsymbol{V}$. $\square$

An interesting observation is that one could show that a strictly block-sparse attention (with $\boldsymbol{V}$ affinely independent) would induce disjoint polytopes by applying the previous lemma to each block, splitting $\mathrm{conv}(\boldsymbol{V})$ as the union of $\mathrm{conv}(\boldsymbol{V}_i)$ for each block $b$. This is particularly interesting as ViT attention patterns are often observed to be block-sparse in practice. This establishes the base case: a single head creates a convex polytope. The next question is: *does this convex structure survive the linear and affine mappings applied throughout the transformer, such as projection matrices or RMSNorm?*

### K.3 ROBUSTNESS TO TRANSFORMATIONS

The first observation is straightforward but crucial: the image of a convex combination under an affine map is the corresponding convex combination of the images. In our setting, this means that if an activation admits an archetypal decomposition, then any affine layer simply moves the archetypes while keeping the codes unchanged. This is exactly the stability we need to propagate archetypal structure through projections and bias terms.

**Lemma 2** (Affine transformations preserve MRH structure). *Let $\gamma(x) = Wx + b$ be an affine transformation and $x = \sum_j z_j a_j$ with $z_j \geq 0$ and $\sum_j z_j = 1$. Then*

$$\gamma(x) = \sum_j z_j a'_j \quad with \quad a'_j = W a_j + b.$$

*Hence convex structure is preserved: archetypes absorbe the transformation, but the codes $z$ remain unchanged.*

*Proof.* Starting from $x = \sum_j z_j a_j$ with $z_j \geq 0$ and $\sum_j z_j = 1$, apply $\gamma$ and use linearity of $W$:

$$\gamma(x) = W\left(\sum_j z_j a_j\right) + b = \sum_j z_j W a_j + b = \sum_j z_j \left(W a_j + b\right) = \sum_j z_j a'_j.$$

The right-hand side is a convex combination of the transformed archetypes $a'_j$. □

Any linear projection $W$ and any bias addition preserve convex decompositions exactly as in Lemma 2. In particular, the per-head output projections and the final output projection of attention blocks do not break archetypal structure. We remark that LayerNorm and RMSNorm are not globally affine because their scale factor depends on the input. However, many norm become affine in evaluation, once mean/variance are held fixed.

### K.4 FROM MULTIPLE HEADS TO MINKOWSKI SUMS

We now turn to the multi-head case. Each head yields a convex polytope (the convex hull of its value vectors), and the standard attention output aggregates head outputs additively after a per-head linear projection. This naturally leads to a Minkowski-sum geometry as describe in Definition 1.

**Proposition 3** (Multi-head attention realizes MRH geometry). *Let there be $H$ heads. For head $h$, let $V_h = \{v_h^{(1)}, \ldots, v_h^{(m_h)}\}$ be its value vectors, and let the head output be*

$$y_h = \sum_{j=1}^{m_h} \alpha_{h,j} v_h^{(j)} \quad with \quad \alpha_h \in \Delta^{m_h}.$$

*Let $W_O^{(h)}$ be the per-head output projection. The block output is*

$$y = \sum_{h=1}^{H} W_O^{(h)} y_h.$$

*Then*

$$y \in \oplus_{h=1}^{H} W_O^{(h)}\big(\mathrm{conv}(V_h)\big),$$

*that is, the attainable outputs lie in the Minkowski sum of the head polytopes after projection. Moreover, under the reachability condition that each head can realize any point of $\mathrm{relint}\big(\Delta^{m_h}\big)$ (up to the softmax additive constant), the set of attainable outputs equals this Minkowski sum. In that case, the representation admits an MRH form with block-convex codes*

$$z = \big(\alpha_1, \ldots, \alpha_H\big) \quad and \ archetypes \quad \mathcal{A} = \big(W_O^{(1)} V_1, \ldots, W_O^{(H)} V_H\big),$$

*where each block $\alpha_h$ belongs to a simplex and each block of archetypes is the projected value set for that head.*

*Proof.* By Lemma 1, for each head $h$ we have $\boldsymbol{y}_h \in \text{conv}(\boldsymbol{V}_h)$. Then, for any linear map $\boldsymbol{L}$ and finite set $S$, we have $\boldsymbol{L}(\text{conv}(S)) = \text{conv}(\boldsymbol{L}(S))$. Therefore $\boldsymbol{W}_O^{(h)}\boldsymbol{y}_h \in \boldsymbol{W}_O^{(h)}(\text{conv}(\boldsymbol{V}_h)) = \text{conv}(\boldsymbol{W}_O^{(h)}\boldsymbol{V}_h)$. By definition of the Minkowski sum, if $\boldsymbol{p}_h \in \boldsymbol{W}_O^{(h)}(\text{conv}(\boldsymbol{V}_h))$ then $\sum_h \boldsymbol{p}_h \in \oplus_h \boldsymbol{W}_O^{(h)}(\text{conv}(\boldsymbol{V}_h))$. Taking $\boldsymbol{p}_h = \boldsymbol{W}_O^{(h)}\boldsymbol{y}_h$ and summing over heads gives

$$\boldsymbol{y} = \sum_{h=1}^{H} \boldsymbol{W}_O^{(h)}\boldsymbol{y}_h \in \oplus_{h=1}^{H} \boldsymbol{W}_O^{(h)}(\text{conv}(\boldsymbol{V}_h)).$$

This proves the inclusion.

Now, assume head $h$ can realize any $\boldsymbol{\alpha}_h \in \text{relint}(\Delta^{m_h})$. Let an arbitrary element of the Minkowski sum be given:

$$\boldsymbol{y}^\star = \sum_{h=1}^{H} \boldsymbol{p}_h \quad \text{with} \quad \boldsymbol{p}_h \in \boldsymbol{W}_O^{(h)}(\text{conv}(\boldsymbol{V}_h)).$$

For each $h$ there exists $\boldsymbol{\beta}_h \in \Delta^{m_h}$ such that $\boldsymbol{p}_h = \boldsymbol{W}_O^{(h)} \sum_j \beta_{h,j} \boldsymbol{v}_{h,j}$. By reachability, choose a query so that the head-$h$ attention equals $\boldsymbol{\alpha}_h = \boldsymbol{\beta}_h$. Then the block output equals $\boldsymbol{y}^\star$. Since $\boldsymbol{y}^\star$ was arbitrary in the Minkowski sum, the attainable set equals the sum.

For the MRH identification, we collect the per-head attention weights into the block vector $\boldsymbol{z} = (\boldsymbol{\alpha}_1, \ldots, \boldsymbol{\alpha}_H)$ and the per-head projected values into archetype blocks $\mathcal{A} = (\boldsymbol{W}_O^{(1)}\boldsymbol{V}_1, \ldots, \boldsymbol{W}_O^{(H)}\boldsymbol{V}_H)$. The output $\boldsymbol{y}$ is thus a sum of $H$ block-convex combinations, realizing the Minkowski sum structure with block-convex coding as required by MRH. $\quad\square$

Having established that attention produces convex polytopes and that affine transformations preserve their structure, we now move toward more realistic operating regimes. What does it mean, geometrically, to have *sparse* or *block-sparse* attention as is commonly observed, and what happens under the elementwise nonlinearities used in practice?

### K.5 ATTENTION CONCENTRATION EFFECTS

We first examine the limit in which softmax sharpens. In fact, as temperature $\tau$ decreases, attention places nearly all mass on the highest-logit index. The attainable set contracts from the full convex hull toward its vertices.

**Lemma 3** (Low-temperature softmax selects vertices)**.** *Consider attention with values $\boldsymbol{V} = \{\boldsymbol{v}_1, \ldots, \boldsymbol{v}_m\}$ and weights $\boldsymbol{\alpha} = \boldsymbol{\sigma}(\boldsymbol{x}/\tau)$ with logits $\boldsymbol{x} \in \mathbb{R}^m$ and temperature $\tau > 0$. Then*

$$\lim_{\tau \to 0} \sum_{j=1}^{m} \alpha_j \boldsymbol{v}_j = \boldsymbol{v}_{j^\star} \quad \text{where} \quad j^\star = \arg\max_j x_j.$$

*In the zero-temperature limit the output lies at a vertex of $\text{conv}(\boldsymbol{V})$.*

*Proof.* For $\tau > 0$, $\alpha_j = \exp(x_j/\tau)/\sum_k \exp(x_k/\tau)$. If $j^\star$ indexes the maximum of $\boldsymbol{x}$ and $\Delta_j = x_{j^\star} - x_j \geq 0$, then $\alpha_{j^\star} = 1/(1 + \sum_{j \neq j^\star} e^{-\Delta_j/\tau})$ and $\sum_{j \neq j^\star} \alpha_j = \sum_{j \neq j^\star} e^{-\Delta_j/\tau} \alpha_{j^\star}$. As $\tau \to 0$, all terms $e^{-\Delta_j/\tau}$ vanish for $\Delta_j > 0$, hence $\alpha_{j^\star} \to 1$ and $\alpha_j \to 0$ for $j \neq j^\star$. The convex combination collapses to $\boldsymbol{v}_{j^\star}$. $\quad\square$

In fact, we can derive a quantitative measure of this convergence, let $\text{diam}(\boldsymbol{V}) = \max_{p,q} \|\boldsymbol{v}_p - \boldsymbol{v}_q\|$ denote the diameter of the value set. Then the deviation from the winning vertex $\boldsymbol{v}_{j^\star}$ satisfies

$$\left\| \sum_j \alpha_j \boldsymbol{v}_j - \boldsymbol{v}_{j^\star} \right\| \leq \text{diam}(\boldsymbol{V}) \sum_{j \neq j^\star} e^{-\Delta_j/\tau},$$

showing that a finite logit margin already forces the output to lie in a small neighborhood of a vertex, with exponentially small deviation in $1/\tau$. Another way to see the effect is to direclty study the geometry created under strict sparsity contraint.

**Lemma 4** (Support restriction selects a subpolytope). *Fix a subset $S \subseteq \{1, \ldots, m\}$ and consider the feasible set of outputs with attention supported on $S$,*

$$\mathcal{Y}_S = \Big\{ \sum_{j \in S} \alpha_j \boldsymbol{v}_j \, : \, \boldsymbol{\alpha} \in \Delta^m, \, \alpha_j = 0 \text{ for } j \notin S \Big\}.$$

*Then $\mathcal{Y}_S = \mathrm{conv}(\{\boldsymbol{v}_j : j \in S\})$. In particular, if across input regimes the support repeatedly takes values in a family $\mathcal{S}$ of subsets, the attainable set is the union $\bigcup_{S \in \mathcal{S}} \mathrm{conv}(\{\boldsymbol{v}_j : j \in S\})$, i.e., a union of lower-dimensional polytopes. When a given $S$ coincides with the maximizers of some linear functional over $\boldsymbol{V}$, $\mathcal{Y}_S$ is a face of $\mathrm{conv}(\boldsymbol{V})$. When $S = \arg\max_j \langle \boldsymbol{w}, \boldsymbol{v}_j \rangle$ for some vector $\boldsymbol{w}$, then $\mathcal{Y}_S$ is the face of $\mathrm{conv}(\boldsymbol{V})$ exposed by the supporting hyperplane with normal $\boldsymbol{w}$.*

*Proof.* Immediate from the definition of convex hull and the constraint $\alpha_j = 0$ for $j \notin S$. The face condition is the standard supporting-hyperplane characterization of faces Rockafellar (1970). $\square$

Lemma 4 formalizes the intuition: as attention sparsifies, geometry collapses from the full polytope to subpolytopes, and in the extreme low-temperature limit to vertices (point).

### K.6 Non-identifiability of Minkowski decomposition

We now address a question that bears directly on recoverability: given only the attainable activation set

$$\mathcal{X} \subset \mathbb{R}^d \quad \text{with} \quad \mathcal{X} = \oplus_{i=1}^m \mathcal{P}_i,$$

can we uniquely recover the summands $\{\mathcal{P}_i\}$? If MRH is to be useful for analysis, we must understand when (and why) such decompositions are non-unique.

**Proposition 4** (**Non-identifiability of Minkowski decomposition**). *Let $\mathcal{X} = \oplus_{i=1}^m \mathcal{P}_i$ be a Minkowski sum of convex polytopes. Given only observations from $\mathcal{X}$, the decomposition $\{\mathcal{P}_1, \ldots, \mathcal{P}_m\}$ is generally non-unique: there exist distinct collections $\{\mathcal{Q}_1, \ldots, \mathcal{Q}_k\}$ such that $\mathcal{X} = \oplus_{j=1}^k \mathcal{Q}_j$. In particular, even very simple polytopes admit infinitely many decompositions as sums of line segments (zonotope generators) with varying directions and lengths.*

*Proof.* We argue via support functions. For a nonempty closed convex set $\mathcal{C} \subset \mathbb{R}^d$, its support function is

$$h_{\mathcal{C}}(\boldsymbol{u}) = \sup_{\boldsymbol{x} \in \mathcal{C}} \langle \boldsymbol{u}, \boldsymbol{x} \rangle \qquad \boldsymbol{u} \in \mathbb{R}^d.$$

Support functions are sublinear (positively homogeneous and subadditive), and they are additive under Minkowski sums Gardner (1995):

$$h_{\mathcal{A} \oplus \mathcal{B}}(\boldsymbol{u}) = h_{\mathcal{A}}(\boldsymbol{u}) + h_{\mathcal{B}}(\boldsymbol{u}) \quad \text{for all } \boldsymbol{u} \in \mathbb{R}^d.$$

Hence a decomposition $\mathcal{X} = \oplus_i \mathcal{P}_i$ is equivalent to a decomposition

$$h_{\mathcal{X}} = \sum_{i=1}^m h_{\mathcal{P}_i}$$

of the single sublinear function $h_{\mathcal{X}}$ into a sum of sublinear summands.

But additive decompositions of a fixed sublinear function are highly non-unique in general. Indeed, fix any sublinear $s$ with $0 \leq s \leq h_{\mathcal{X}}$ pointwise. Then both

$$h_{\mathcal{X}} = (h_{\mathcal{X}} - s) + s \quad \text{and, more generally, } \forall \{h_1, ..., h_m\} \text{ s.t. } \sum_{i=1}^m h_i = h_{\mathcal{X}}$$

all define a valid support-function decompositions. Under lower semicontinuity, each sublinear $h_i$ is itself the support function of a unique closed convex set $\mathcal{Q}_i$ containing the origin, i.e., $h_{\mathcal{Q}_i} = h_i$. Therefore

$$h_{\mathcal{X}} = \sum_{i=1}^m h_i \quad \Longleftrightarrow \quad \mathcal{X} = \oplus_{i=1}^m \mathcal{Q}_i.$$

Since there are infinitely many ways to split $h_{\mathcal{X}}$ into a sum of sublinear functions, there are infinitely many corresponding Minkowski decompositions. This proves the general non-uniqueness claim. $\square$

To build concrete intuition on why non-uniqueness appear, consider the simplest case of decomposing a rectangle as a sum of segments.

**Segments generating the same zonotope** We work in $\mathbb{R}^2$ and consider the axis-aligned rectangle

$$\mathcal{X} = [-a, a] \times [-b, b] = ([-a, a]\mathbf{e}_1) \oplus ([-b, b]\mathbf{e}_2).$$

For any $\alpha \in (0, 1)$,

$$[-a, a]\mathbf{e}_1 = [-\alpha a, \alpha a]\mathbf{e}_1 \oplus [-(1-\alpha)a, (1-\alpha)a]\mathbf{e}_1,$$

so

$$\mathcal{X} = [-\alpha a, \alpha a]\mathbf{e}_1 \ \oplus \ [-(1-\alpha)a, (1-\alpha)a]\mathbf{e}_1 \ \oplus \ [-b, b]\mathbf{e}_2.$$

Varying $\alpha$ yields uncountably many distinct 3-segment decompositions of the same rectangle. Also, iterating this splitting on either axis leads to infinitely many distinct $k$-segment decompositions whose projections sum to the same total width and height. Hence even for very simple polytopes, Minkowski-sum decompositions into segments are non-unique.

## K.7 EMPIRICAL EVIDENCE

We now detailed for the preliminary evidence exposed in Section 6, for each of the 3 criterion of the hypothesis. The results below should be read as compatible evidence, not proof: multiple mechanisms can mimic the same surface phenomena. Unless stated otherwise we use ImageNet-1k validation tokens from the last DINOv2-B layer; cosine distance and $k$-NN graphs with standard symmetrization.

Figure 26 (left) contrasts straight-line interpolation between tokens with piecewise-linear geodesics computed as shortest paths on the token $k$-NN graph. Straight lines depart the data support quickly, whereas graph geodesics remain close throughout. This matches what a Minkowski sum structure predicts: feasible displacements arise as sums of small face-walks within head polytopes, which are piecewise linear in barycentric coordinates yet appear curved in the ambient space. The curved, on-support geodesics thus support criterion (*i*) by reflecting head-wise convex reweighting rather than simple linear combinations.

To test the convex coding assumption (*ii*), we compare Archetypal Analysis Cutler & Breiman (1994) (AA) with SAE for token reconstruction in Figure 26 (middle). Note that AA is precisely the single-tile case of MRH ($|S| = 1$), making this a direct test of our geometric assumptions. AA imposes dramatically stronger constraints[3]: it forces all reconstructions to lie within the convex hull of observed tokens and requires archetypes to be actual data combinations. Despite these restrictive assumptions, AA matches SAE performance with remarkably few archetypes (10 archetypes per image), providing preliminary evidence that even the simplest case of MRH captures fundamental geometric structure.

This is particularly striking in high dimensions, where the probability that a random point lies in the convex hull of a small set of samples decays exponentially with dimension Bárány & Füredi (1988); Balestriero et al. (2021); the fact that tokens can nevertheless be accurately reconstructed from only ∼10 archetypes indicates that embeddings concentrate on low-dimensional polytopes embedded within the ambient space.

Additionally, the block structure emerges naturally: Figure 26 (right) reveals that archetypal decompositions spontaneously organize into the block-sparse pattern predicted by criterion (*iii*), with distinct clusters of co-activating archetypes rather than uniform mixing.

We emphasize that these observations – piece-wise linear on-manifold trajectories, efficient archetypal reconstructions, and block-structured co-activation – represent preliminary evidence rather than definitive proof. Multiple geometric hypotheses could generate similar surface patterns. Nevertheless, the convergent evidence motivates exploring what MRH would mean for interpretability practice.

**Positioning & Claims.** We treat the Minkowski Representation Hypothesis (MRH) as a working hypothesis, not a proved theory of representation. We do not claim causal identification of concepts, head "tiles," or their generative mechanisms; our evidence is observational and model-specific (DINOv2-B). Our goal is to describe robust empirical regularities (task-specific subspaces; families

---

[3]Archetypal Analysis seeks $Z = XB$ where archetypes are convex combinations of data (so $Z \subset \text{conv}(X)$), and $X \approx ZA$ where data are convex combinations of archetypes. Both $A$ and $B$ have simplex constraints (columns sum to 1, non-negative).

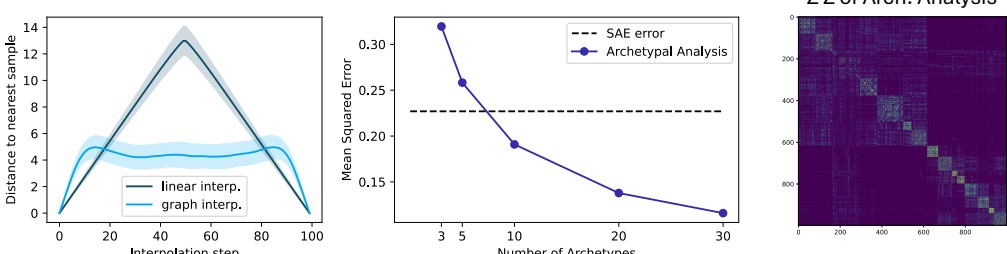

Figure 26: **Empirical support for MRH criteria on ImageNet-1k validation set. (Left)** Distance to data along interpolation paths between tokens. Linear interpolation (dark line) rapidly deviates from valid embeddings, while there exist piecewise-linear paths computed via shortest paths on token $k$-NN graphs (light blue) that remain consistently close to the data manifold. This supports criterion (***i***) Minkowski sum structure: feasible connections follow polytope face-walks rather than straight lines through empty space. **(Middle)** Archetypal Analysis (single-tile MRH with $|S| = 1$) achieves lower error than a sparse autoencoder (dashed line) with as few as 10 archetypes, despite stronger constraints, supporting criterion (***ii***) convex coding assumptions. **(Right)** Archetypal coefficient matrix $\boldsymbol{Z}^T \boldsymbol{Z}$ after clustering reveals emergent block structure with bright diagonal clusters. Even without knowledge of tile boundaries, archetypes naturally organize into co-activating groups, supporting criterion (***iii***) block-structured decomposition.

of depth cues; "Elsewhere" patterns) and propose a geometry that makes testable predictions for future work (e.g., per-head block structure, subspace additivity across heads). We therefore refrain from stating that our results "contradict" LRH; instead, we document systematic departures from a purely sparse near-orthogonal picture. All conclusions should be read as conditional on our dictionary, probes, and datasets, and we report ablations where feasible.

## L    VISUALIZATION TOOL

To facilitate exploration of the concept dictionary and make our findings accessible to the broader research community, we release **Dino**Vision, an interactive web-based visualization tool that enables real-time exploration of the 32,000 extracted concepts. The tool presents concepts as points in a 2D UMAP projection where spatial proximity indicates conceptual similarity in the original high-dimensional space, though global clustering patterns should be interpreted with caution due to UMAP's limitations in preserving large-scale structure. The interface displays each concept as a

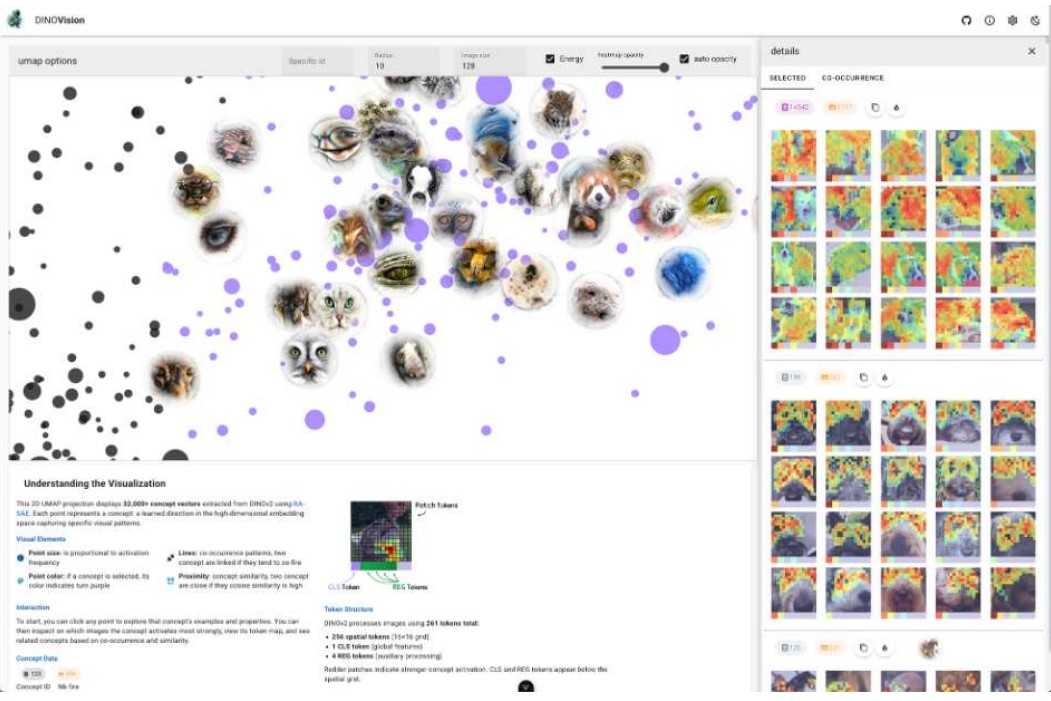

Figure 27: DinoVision interface showing the interactive 2D UMAP projection of 32,000 concepts extracted from DINOv2. Users can explore individual concepts by clicking points to reveal activation patterns across the 261 token grid (256 spatial patches plus CLS and register tokens). The visualization includes adjustable parameters for point size, opacity, and co-occurrence links between frequently activated concept pairs.

point whose size reflects activation frequency across the dataset. Users can click any point to examine detailed activation patterns showing how that concept fires across DINOv2's 261 tokens, which comprise 256 spatial patches arranged in a $16 \times 16$ grid plus one classification token and four register tokens. The token visualization uses color intensity to indicate activation strength, with redder regions corresponding to stronger concept responses. The tool help us discovered that some concepts activate exclusively on register tokens, and they seems to capturing global scene properties like illumination and motion blur. Interactive features include adjustable visualization parameters such as point size scaling, heatmap opacity controls, and the ability to display co-occurrence links between concepts that frequently activate together. Users can navigate directly to specific concepts by entering concept identifiers or explore neighborhoods around selected points. The co-occurrence analysis reveals structured relationships in the concept space, with connecting lines indicating statistical dependencies between concept activations. For concept visualization we use Feature Accentuation (FA) from Hamblin et al. (2024), we start from maximally activating images of each concept and perform 1024 steps of gradient ascent optimization parameterized in Fourier space with MACO Fel et al. (2023a) constraints on the magnitude of the spectrum, boosted according to natural image statistics approximately following $1/\omega^2$ where $\omega$ represents cycles per image.

Importantly, the tool implements a composite (two-layer) rendering approach that maintains smooth 60fps interaction for point navigation while progressively loading high-resolution concept visualiza-

tions as needed. We believe the tool serves as both a research instrument for further investigation and a demonstration of the rich structure present in vision transformer representations.

