# OpenReview forum: "Into the Rabbit Hull: From Task-Relevant Concepts in DINO to Minkowski Geometry"
_ICLR.cc/2026/Conference — ICLR 2026 Poster_

### Official Review · Reviewer_vpmQ · 2025-10-20

**Soundness:** 2
**Presentation:** 3
**Contribution:** 4
**Rating:** 8
**Confidence:** 3

**Summary:**

Using a stable sparse autoencoder, the paper builds a large concept dictionary for DINOv2 and shows task-specific feature usage. Analyses of concept statistics and geometry reveal distributed, partially sparse structure that only partially matches the Linear Representation Hypothesis. Finally, the authors propose the Minkowski Representation Hypothesis: token embeddings lie in sums of convex regions (aligned with multi-head attention’s convex mixing), which have implications in mechanistic interpretability such as model steering.

**Strengths:**

Clarity:
- The paper follows a mostly logical narrative structure: Testing LRH with Stable SAE -> Empirical findings on concept statistics and geometry -> MRH
- Once released, the interactive concept explorer can make the work much easier to visualize.

Quality:
- The methodology is rigorous, relying on quantitative measures for its major claims (singular-value spectra, coherence vs Grassmannian/random baselines, etc).
- The convex-hull constraint on the SAE makes it more reproducible by keeping concepts in-distribution.

Originality:
- While concepts such as superposition and polysemanticity have been widely explored in mechanistic interpretability, the idea of convex regions is quite novel in this area, to the best of my knowledge.

**Weaknesses:**

l161: Because the stable SAE constrains dictionary atoms to the convex hull of activations, convexity is guaranteed for the learned concepts but not necessarily for the native representation. Maybe explicitly separate SAE-induced claims from “model-native” claims, and add ablations with unconstrained SAE, showing the same qualitative geometry without the convex prior.

l064: The importance definition is clear, but the figure mapping isn’t. Maybe specify exactly what dot size and color encode in Figure 1. Adding a concise legend in the main text, a one-line formula linking $\phi$ to the plotted quantity would make the visualization reproducible and easier to see.

l259: UMAP is fine for intuition, but it shouldn’t be used to make claims. Maybe caution that UMAP as visualization only, add a brief hyper-parameter sensitivity (n_neighbors, min_dist, seed) in the appendix, and keep the main analysis on PCA/spectral metrics.

l293: "dense Sun et al. (2025) activations" -> "dense activations Sun et al. (2025)"

**Questions:**

See weaknesses.

---

> ### Author Response · Authors · 2025-11-18
> **Response**
>
> **We thank you for the strong endorsement** and recognition of our narrative clarity, quantitative rigor, and the novelty of the convex-region perspective. Below, we address each of your points.
>
> ---
>
> _“l161: Because the stable SAE constrains dictionary atoms to the convex hull of activations, convexity is guaranteed for the learned concepts but not necessarily for the native representation. Maybe explicitly separate SAE-induced claims from “model-native” claims, and add ablations with unconstrained SAE, showing the same qualitative geometry without the convex prior.”_
>
> This is an important distinction that we will state explicitly. While the stable SAE constrains dictionary atoms to the convex hull, **our MRH tests operate directly on native DINOv2 tokens without this constraint**.
>
> _l”064: The importance definition is clear, but the figure mapping isn’t. Maybe specify exactly what dot size and color encode in Figure 1. Adding a concise legend in the main text, a one-line formula linking to the plotted quantity would make the visualization reproducible and easier to see.”_
>
> **Yes, you are correct**, and we will add a comprehensive legend with the associated formula to make the visualization easier to grasp.
>
> _“l259: UMAP is fine for intuition, but it shouldn’t be used to make claims. Maybe caution that UMAP as visualization only, add a brief hyper-parameter sensitivity (n_neighbors, min_dist, seed) in the appendix, and keep the main analysis on PCA/spectral metrics.”_
>
> We fully agree with the reviewer. We only use UMAP once (Fig. 1) as an illustrative visualization to convey the intuitive clustering of task-relevant concepts. **All quantitative claims about low-dimensionality are based on quantitative measure: specifically, the spectral decay of singular values in the task-restricted sub-dictionaries (Fig. 11 right). We did not draw any inferential conclusions from UMAP.**
>
> _l293: "dense Sun et al. (2025) activations" -> "dense activations Sun et al. (2025)"_
>
> Thank you for catching this !
>
> ---
>
> Thank you for your feedback on these methodological details!

---

### Official Review · Reviewer_PH2n · 2025-10-27

**Soundness:** 3
**Presentation:** 4
**Contribution:** 2
**Rating:** 4
**Confidence:** 3

**Summary:**

This paper introduces a 32k-concept dictionary extracted from DINOv2 using Sparse Autoencoders (SAEs) and presents an interactive web-based visualization platform. The authors analyze how downstream tasks selectively recruit concepts from this dictionary, finding that classification relies on "Elsewhere" concepts implementing object negation, segmentation uses boundary detectors forming coherent subspaces, and depth estimation draws on three families of monocular cues (projective geometry, shadows, and frequency transitions). The work proposes the Minkowski Representation Hypothesis (MRH) as an alternative to the Linear Representation Hypothesis (LRH), suggesting that token embeddings behave as sums of convex regions around archetypal landmarks rather than as sparse combinations of near-orthogonal directions. However, the authors acknowledge that MRH remains a working hypothesis requiring further empirical validation.

**Strengths:**

**Clear introduction and methodological foundation**: The paper provides adequate background on ViTs, DINOv2, vision explainability, SAE adoption, and task-specific learned concepts, making the work accessible to the broader audience.

**Interpretability contributions**: The interpretation sections are particularly well-executed, and the release of a web-based visualization platform for navigating DINOv2's 32k-concept dictionary represents a valuable resource for the community.

**Technical rigor**: The submission is mostly clear, technically correct, and results appear reproducible given the extended mathematical explanations, despite code not being provided.

**Novel theoretical perspective**: The paper proposes MRH as an interpretability framework for understanding ViTs, suggesting future research should examine these models through this geometric lens.

**Weaknesses:**

**Limited scope of downstream task exploration**: Section 3 examines only three downstream tasks, while DINOv2 supports many more applications. For classification, only the "elsewhere" concept is explored in detail, limiting the generalizability of findings.

**Insufficient empirical foundation for MRH**: The authors propose MRH based on intuitions from task-specific geometrical organization but acknowledge it still needs rigorous proof (line 445). Given the limited number of tasks and task-specific concepts explored (see point above), the theoretical proposition rests on unstable empirical grounds.

**Underdeveloped discussion of implications**: The paper lacks sufficient explanation of what consequences adopting the MRH concept has for interpreting ViTs and what concrete benefits this perspective provides to the research community. The discussion section should match the verbosity and clarity of the introduction.

**Minor:**
- Misleading characterization of SAE contribution: The authors state they "operationalize" LRH using SAE, when they actually adopt an SAE previously introduced by Fel et al. (2025). This overstates their methodological contribution.
- Excessive reliance on appendix: Figures in the appendix are cited frequently in the main text (e.g., "Different tasks recruit different concepts" in Sec. 3, "Empirical evidences" in Sec. 6), resulting in sections that are difficult to follow without constant back-and-forth navigation.
- Citations should stay in brackets, use \citet.
- TL;DR is broken.

**Questions:**

- Sec. 3: what's the reason for isolating the top 100 most task-aligned concepts per-head when analyzing their similarities?
- Sec. 3: how do you defined the top concepts for classification and how important or frequent is the "elsewhere" concept? why only this concept is analyzed for the classification task?
- Sec. 3: you cite Fig.10 (right) in appendix but the same can be seen in Fig. 2
- Is "the largest interactive interpretability demo" (line 112) an advantage? isn't complexity a problem for interpretability?
- What's the x-axis of Fig.11 left?

---

> ### Author Response · Authors · 2025-11-18
> **Response -- Part 1**
>
> Thank you for the fair and precise review, recognizing our clear methodology, interpretability contributions, and the value of our interactive explorer. We have taken this review seriously and respond to specific comments below.
>
> ---
>
> _“Limited scope of downstream task exploration: Section 3 examines only three downstream tasks, while DINOv2 supports many more applications. [...]”_
>
> We acknowledge this limitation while noting **we are, to our knowledge, the first work to apply SAE-based interpretability to the 3 tasks of depth estimation, classification and segmentation** as most prior work focuses solely on classification. We selected these three tasks as they represent DINO's main use cases. That said, we're genuinely curious: what additional tasks would the reviewer suggest? We're ready to add another task if it would strengthen the analysis. We chose our three tasks believing they capture how practitioners typically use DINO, but we may be wrong about usage patterns.
>
> _“Insufficient empirical foundation for MRH: The authors propose MRH based on intuitions from task-specific geometrical organization but acknowledge it still needs rigorous proof (line 445). Given the limited number of tasks and task-specific concepts explored (see point above), the theoretical proposition rests on unstable empirical grounds.”_
>
> We respectfully disagree with this assessment and believe there may be a misunderstanding of our paper's structure. We never claim MRH emerges from task-specific geometric patterns, we're unsure where the reviewer sees this suggested.
> Our development follows this path: after observing deviations from pure LRH (Section 4), we returned to first principles and analyzed the mathematical structure of multi-head attention. This theoretical analysis reveals that attention heads necessarily produce convex combinations of their values (Lemma 1), and when multiple heads sum their outputs, this creates a Minkowski sum of convex polytopes (Proposition 3). **It is not an intuition; it is simply the geometry that multi-head attention mechanically produces**.
>
> **The empirical evidence in Section 6 and Figure 26 serves to then validate these theoretical predictions**, not to motivate them. We provide three quantitative tests: (1) geodesic paths remain closer to the data manifold via piecewise-linear trajectories than paths following standard linear interpolation, (2) Archetypal Analysis achieving comparable performance to SAEs despite enforcing convexity constraints, and (3) emergent block structure in activation patterns matching our theoretical predictions implied by the tiles.
>
>
> _“Underdeveloped discussion of implications: The paper lacks sufficient explanation of what consequences adopting the MRH concept has for interpreting ViTs and what concrete benefits this perspective provides to the research community. The discussion section should match the verbosity and clarity of the introduction.”_
>
> We're puzzled by this critique, as we have a dedicated section in the main paper explicitly addressing MRH's implications (e.g., L445). We recall here the three concrete consequences if MRH holds:
> - First, concepts are regions (delimited by points that we call archetypes), not unbounded directions; this fundamentally changes how we should interpret and look for concepts.
> - Second, this implies that steering has natural saturation limits since moving toward archetypal landmarks eventually reaches polytope boundaries, that could explain why SAE-based steering often plateaus or reverses at high magnitudes.
> - Third, decomposition is non-identifiable from final activations alone (Proposition 2), meaning we need intermediate signals (e.g., attention weights, Intermediates values of attention block, etc) for proper identifiability.
>
> _“Misleading characterization of SAE contribution: The authors state they "operationalize" LRH using SAE, when they actually adopt an SAE previously introduced by Fel et al. (2025). This overstates their methodological contribution.”_
>
> You are right, our phrasing could be misunderstood. The SAE we use is not the contribution of our current paper. We did not intend to claim methodological novelty here, but rather **to make explicit the link between representational assumptions and appropriate methods.**
> Our point is that operationalizing LRH naturally leads to SAEs as the extraction method. **If one changes the underlying geometric assumption, the method should change accordingly. This duality between hypothesis and method is why we emphasized the LRH-to-SAE connection at the outset.**
>
> The transition to MRH follows this logic. It motivates a different operational lens for interpretability rather than a different training procedure. We will clarify this in revision to avoid misunderstanding.

---

> ### Author Response · Authors · 2025-11-18
> **Response -- Part 2**
>
> _“Excessive reliance on appendix: Figures in the appendix are cited frequently in the main text (e.g., "Different tasks recruit different concepts" in Sec. 3, "Empirical evidences" in Sec. 6), resulting in sections that are difficult to follow without constant back-and-forth navigation.”_
>
> **We sincerely apologize for this**. We would like to thank the reviewer one more time for their comments and patience. Furthermore, we recognize that asking reviewers to parse such a submission could be a substantial request.
>
> _“Citations should stay in brackets, use \citet. TL;DR is broken.”_
>
> We are grateful for your sharp eye and attention to detail regarding the manuscript's presentation. We will address all formatting issues: fixing citations to use \citet, repairing the TL;DR, moving key figures from appendix to main text, and adding explicit formulas for all visualization mappings. Thank you again for all those comments that will improve the paper.
>
> _“Sec. 3: what's the reason for isolating the top 100 most task-aligned concepts per-head when analyzing their similarities?”_
>
> We applied a cutoff to normalize comparisons across heads with different value dimensions. The qualitative and quantitative patterns remain unchanged when selecting the top-10, top-100, or top-500 most task-aligned concepts. **In fact, the structure becomes even clearer with top-10 selections**.
>
> _“Sec. 3: how do you defined the top concepts for classification and how important or frequent is the "elsewhere" concept? why only this concept is analyzed for the classification task?”_
>
> Top concepts are ranked by the importance score defined in Appendix C.1 (the amount of energy a concept sends into the logit). Elsewhere concepts are defined as concepts whose token activation map is spatially the negative mask with attribution: they fire strongly off-object while attribution highlights the object itself.
>
> These concepts consistently rank in the top 3 for every ImageNet class we examined. We have not found a class without an associated Elsewhere concept. **We focus on them because they represent a systematic, previously undocumented pattern across classification, not an isolated phenomenon.**
>
> _“Sec. 3: you cite Fig.10 (right) in appendix but the same can be seen in Fig. 2”_
>
> Thank you for catching this. We will correct the reference to cite Fig 2 instead.
>
>
> _“Is "the largest interactive interpretability demo" (line 112) an advantage? isn't complexity a problem for interpretability?”_
>
> **Fair point! In fact, the authors agree with you that we shouldn’t claim that scale alone is beneficial, and our phrasing was suggesting that**. To clarify our perspective, following Edgar Morin's principle of _‘how to embrace complexity in a non-reductive manner?’_, we believe complexity is not inherently the enemy of understanding. The challenge is navigating complexity without reductive oversimplification (e.g. scale would be beneficial if it facilitates somehow the navigation of the representation space). Does this framing seem reasonable to you?
>
> Either way, thanks for raising this, it sparked valuable discussion among the authors, and we believe the tension between comprehensiveness and interpretability in interactive demo is underdiscussed in the field and worth emphasizing in the manuscript.
>
> _“What's the x-axis of Fig.11 left?”_
>
> Good catch, sorry for that, the x-axis shows importance score as defined in Appendix C.1, and the y-axis shows density. The plot displays how many concepts have each level of importance for each task. We will add explicit axis labels to clarify this.
>
> ---
>
> Thank you for this thorough and constructive review. While we respectfully disagree on a few points (we believe MRH emerged from first-principles attention analysis) **we found your feedback genuinely helpful and well-articulated**. We believe your feedback has strengthened the manuscript, and we appreciate the time and care you invested in engaging with our work.

---

### Official Review · Reviewer_F23h · 2025-11-01

**Soundness:** 4
**Presentation:** 4
**Contribution:** 4
**Rating:** 8
**Confidence:** 2

**Summary:**

In this work, the authors analyze the internal representations of the DinoV2 model to provide interpretability insights into how the model performs tasks. They begin by characterizing tokens using a sparse autoencoder to create a concept dictionary. They then investigate how different tasks use this dictionary to accomplish their distinct goals and highlight their distinct differences. For example, they find that classification uses elsewhere concepts while segmentation uses boundary detectors. Secondly, the authors find that the representations are partially dense demonstrating behavior that may be inconsistent with sparse coding. Finally, the authors propose a new hypothesis called the Minkowski Representation Hypothesis which they suggest as an alternative to the LRH. They provide initial evidence for the hypothesis, then leave room for investigations in the future.

**Strengths:**

This paper demonstrates clear novelty and depth while providing a useful tool for other researchers to investigate further. These qualities constitute a clear accept.

**Weaknesses:**

The analysis in this paper is mostly descriptive as opposed to describing why different tasks within DINOv2 have different representational sparseness etc.

**Questions:**

One weakness of this paper is that it primarily is descriptive as opposed to providing an understanding of why each task has such different utilization of concepts. Can the authors describe why they believe this asymmetry is happening? Is it relative complexity between the task types etc?

The authors suggest that Dinov2’s representations follow their proposed Minkowski Representation Hypothesis. One thing that isn’t clear is whether the authors believe that other networks should follow this hypothesis. It seems to stem very naturally from attention layers. Do the authors expect other networks to follow this hypothesis? If so, when do they expect it to emerge over LRH or do they consider it an alternative hypothesis to LRH in general?

---

> ### Author Response · Authors · 2025-11-18
> **Response**
>
> **Thank you for your strong endorsement** and appreciation of our novelty, depth, and the utility of our visualization tool. Below we address specific comments.
>
>
> _“One weakness  [...]primarily is descriptive as opposed to providing an understanding of why each task has such different utilization of concepts. Can the authors describe why they believe this asymmetry is happening? Is it relative complexity between the task types etc?”_
>
> **This is an excellent question**. Honestly, we don't know why. We observe clear empirical patterns but the underlying mechanisms remain unclear. Several hypotheses seem plausible. Probe rank may matter: classification requires a higher-dimensional decision boundary (cf importance density plot) than segmentation or depth, naturally recruiting more concepts. Task supervision could shape recruitment: pixel-level labels for segmentation versus image-level labels for classification may induce different functional pressures. Also, as you pointed out, intrinsic task complexity likely differs: recognizing object categories requires integrating diverse evidence while detecting boundaries relies on localized edge information.
>
> These are speculations, not conclusions supported by our experiments. What we can state confidently is that the differences are real, substantial, and consistent across our analyses.
>
> We also agree this is primarily descriptive work. Understanding why these patterns emerge is an important direction for future research. **We believe documenting the phenomenon rigorously is a necessary first step toward mechanistic explanations.**
>
>
> _“The authors suggest that Dinov2’s representations follow their proposed Minkowski Representation Hypothesis. [...]Do the authors expect other networks to follow this hypothesis? If so, when do they expect it to emerge over LRH or do they consider it an alternative hypothesis to LRH in general?”_
>
> Thank you for this comment. In this work we tried to specifically avoid overclaiming on this question as we believe establishing whether MRH describes a general phenomenon would require substantially more experimentation across architectures, training procedures, and domains.
>
> However, we can make a more limited claim with reasonable confidence. Because multi-head attention with standard softmax inherently produces convex combinations per head, as proven in Lemma 1, and sums these combinations across heads as shown in Proposition 3, we expect MRH-style geometry to potentially emerge in vision transformers using non-causal attention.
>
> Regarding the relationship to LRH, we view them as complementary rather than contradictory. LRH provides a valid local linear approximation near polytope faces where small perturbations behave linearly. MRH describes the global manifold structure governing how these local regions connect.
>
> ---
>
> Thank you again for the thoughtful questions, your point about task-specific mechanisms is something we'll definitely explore further. We appreciate you taking the time to engage seriously with the work, and we're excited to see where this leads next.

---

### Official Review · Reviewer_nELv · 2025-11-04

**Soundness:** 3
**Presentation:** 3
**Contribution:** 3
**Rating:** 6
**Confidence:** 3

**Summary:**

The paper studies what DINOv2 “sees” by operationalizing the Linear Representation Hypothesis (LRH) with a sparse autoencoder to extract a 32k-concept dictionary from ViT activations, then analyzing how downstream tasks recruit these concepts and what the concepts’ geometry suggests. Empirically, different tasks use distinct, low-dimensional subsets: classification leans on off-object “Elsewhere” concepts, segmentation concentrates on boundary detectors, and depth estimation draws on monocular cue families. they also run geometric diagnostics and find some deviations from the LRH like structured redundancy. Motivated by this, the authors propose the Minkowski Representation Hypothesis (MRH): token embeddings lie in Minkowski sums of convex polytopes spanned by archetypal landmarks, a structure they argue multi-head attention realizes via convex combinations per head and summation across heads. They provide preliminary qualitative/quantitative signals consistent with MRH - e.g. token embeddings smoothly interpolate between landmark-like prototypes instead of varying along linear feature axes.

**Strengths:**

- The paper is ambitious in scope, undertaking one of the first large-scale interpretability analyses of a state-of-the-art vision foundation model through SAEs.
- Goes beyond static visualization to quantify how distinct downstream tasks (classification, segmentation, depth) selectively recruit different subsets of the concept space, revealing functional specialization. e.g. the paper identifies interpretable and generalizable phenomena—e.g., “Elsewhere” (off-object) concepts, border detectors, and monocular depth cues—illustrating that meaningful, task-specific structure emerges spontaneously in DINO.
- Moves beyond the traditional “sparse direction” view of representations to analyze the geometric structure (anisotropy, coherence, spectra) of learned concepts
- The work also makes a creative theoretical leap—proposing the Minkowski Representation Hypothesis to reinterpret transformer geometry in terms of convex archetypal regions, an idea that elegantly ties attention mechanics to cognitive theories of conceptual spaces.

**Weaknesses:**

- The paper attempts to do too much—spanning SAE implementation, large-scale task analysis, and a new geometric theory (MRH)—without a unifying throughline. The connection between these parts often feels narrative rather than logically necessary.
- The evidence for the Minkowski Representation Hypothesis is largely qualitative and circumstantial (e.g., UMAPs, smooth PCA maps, block structures). Stronger quantitative tests or falsifiable predictions are needed to substantiate the claim.
- The observed deviations from LRH (anisotropy, coherence, low-dimensional task subspaces) could be explained by simpler mechanisms such as structured sparsity or normalization effects—yet the paper moves quickly to a new geometric framework without ruling these out.
- Key hyperparameters (e.g., SAE sparsity level, dataset etc.) and their sensitivity are not well explored; it is unclear how robust the reported concept structure for the 3 tasks is to these design choices.
-  While task specialization is interesting, it remains correlational; it’s unclear whether manipulating the discovered concepts truly affects task behavior. I think some causal manipulation could be interesting too.

**Questions:**

- How do you quantitatively assess convex regions or Minkowski sums beyond analogy to attention mechanics?
- How sensitive are the learned concept dictionaries to SAE initialization and regularization strength?
- How do the discovered “Elsewhere” concepts improve human interpretability or model steering compared to prior SAE- or NMF-based approaches?

---

> ### Author Response · Authors · 2025-11-18
> **Response - Part 1**
>
> **Thank you for your thoughtful and constructive review!**
>
> We appreciate your recognition of the paper’s strengths, including its “ambitious” scope and the “creativity” of the representation hypothesis. Below, we address each of your points in detail.
>
> ---
>
> _“The paper attempts to do too much [...] without a unifying throughline. The connection between these parts often feels narrative rather than logically necessary.”_
>
> We acknowledge this important structural feedback and will strengthen the logical connections in the revision. **We will explicitly state upfront our progression**: (1) we begin with the Linear Representation Hypothesis as our operational baseline via stable SAE, (2) quantify how different downstream tasks recruit concepts revealing distinct functional specializations, (3) document systematic deviations from LRH that point towards more structure than pure feature packing, and (4) propose a refined geometric account grounded in attention mechanics.
>
> We will add clear signposts throughout the manuscript to make this logical flow more apparent.
>
>
> _“The evidence for the Minkowski Representation Hypothesis is largely qualitative and circumstantial (e.g., UMAPs, smooth PCA maps, block structures) [...].”_
>
> We respectfully disagree with this characterization. **Our MRH is not based on UMAP or PCA visualizations**. In fact, we use UMAP once in Figure 1 for task recruitment, and we draw no geometric conclusions from it. Even the claim about low-dimensional task subspaces comes from spectral analysis (and has nothing to do with MRH).
>
> Regarding MRH, it rests on architectural mechanics proven in Proposition 1 and Lemmas 1-2. Single attention heads produce convex combinations of values. Affine projections preserve this structure. Multi-head summation yields Minkowski sums. These follow from the transformer design. Furthermore, figure 26 provides quantitative falsifiable tests. Geodesic paths stay on-manifold while Euclidean paths leave it. Archetypal Analysis with 10 archetypes per image matches SAE despite strict convexity constraints. The Gram matrix Z transpose Z exhibits block structure, not random co-activation.
>
> However, **we agree with you (and we state explicitly at line 445) on the fact that MRH is a working hypothesis**. The theoretical foundation is rigorous, but the empirical tests are preliminary. We believe the idea of concepts as convex-regions is underexplored and merits exploration as an alternative to purely sparse-coding frameworks.
>
> _“The observed deviations from LRH [...]could be explained by simpler mechanisms [...].”_
>
> **We appreciate this point**. Multiple hypotheses could explain our observations. The reviewer suggests structured sparsity and normalization effects as simpler mechanisms. Interestingly, these are precisely what MRH formalizes: structured sparsity is tile-based block activation, and normalization is the convex constraint on combinations within each tile.
> More generally, one could augment LRH with such constraints post hoc to account for the patterns we document. We chose a different path guided by simplicity. Rather than adding complexity to LRH, we examined attention's mechanics directly and MRH emerges from what attention already does: convex combinations per head, summation across heads. No additional assumptions required.
>
> We tested alternatives where feasible. Appendix J controls for position via orthogonal projection. Appendix F compares against random and Grassmannian baselines. Our goal is not to definitively disprove augmented LRH variants. We believe MRH offers a complementary geometric framework worth investigating.
>
> Thanks for this comment and letting us clarify our position!

---

> > ### Author Response · Authors · 2025-11-18
> > **Response - Part 2**
> >
> > _“Key hyperparameters (e.g., SAE sparsity level, dataset etc.) and their sensitivity are not well explored; [...]”_
> >
> > We deliberately chose the stable RA-SAE for reproducibility, as the results are largely independent of the seed. Regarding the single method, **we believe that interpretability can also benefit from deeply studying one configuration rather than shallow sweeps across many**. This choice enabled discoveries like 'Elsewhere' concepts and monocular depth cue families that required sustained qualitative investigation.
> >
> > We verified key findings (especially 'Elsewhere' concepts) also emerge with standard SAEs, suggesting these reflect genuine model structure rather than method artifacts. While we acknowledge future work should explore sparsity levels (k) and dictionary sizes systematically, we prioritized depth over breadth (uncovering rich phenomena through careful inspection rather than extensive ablations).
> >
> > _“While task specialization is interesting, it remains correlational; it’s unclear whether manipulating the discovered concepts truly affects task behavior., [...] some causal manipulation could be interesting too.”_
> >
> > We respectfully clarify that our task-concept alignment is already causal, not correlational. As detailed in Appendix C.1, our importance measure directly quantifies each concept's causal contribution to task logits (we fuse the dictionary and the linear probe weights and that allows us to directly quantify the amount of energy that each concept sends into the logit). Removing high-importance concepts would mechanistically remove the largest logit contributions, making this a causal relationship by construction.
> >
> > Thank you for the remark, we will expand and clarify that aspect in the manuscript.
> >
> >
> > _“How do you quantitatively assess convex regions or Minkowski sums beyond analogy to attention mechanics?”_
> >
> > **This is a good question**, and the answer is: it's not easy. We report three complementary preliminary results in that direction, presented in Figure 26.
> > - First, geodesic analysis. If tokens lie in Minkowski sums of polytopes, moving between tokens should follow piecewise-linear trajectories along polytope faces. We find that paths through the token k-NN graph remain consistently closer to observed data than straight Euclidean interpolations, suggesting the manifold has piecewise-linear rather than globally-linear structure.
> > - Second, Archetypal Analysis reconstruction. AA enforces strict convexity: archetypes are convex combinations of data, data are convex combinations of archetypes. This is a single-tile MRH. If tokens did not concentrate on convex structures, AA would fail in high dimensions. Instead, it matches SAE with only 10 archetypes per image.
> > - Third, block-Gram analysis. MRH predicts block-sparse co-activation aligned with tiles. Z transpose Z shows distinct diagonal blocks where archetypes activate together, not uniform or random patterns.
> >
> >
> > _“How do the discovered “Elsewhere” concepts improve human interpretability or model steering compared to prior SAE- or NMF-based approaches?”_
> >
> > **We believe the primary goal of interpretability is to discover and characterize what models actually do, regardless of whether the mechanisms are intuitive or convenient for human understanding**. Elsewhere concepts are a discovery: they reveal that DINOv2 systematically uses spatially inverted features for classification, where concepts fire strongly in regions that do not contain the object they help detect.
> >
> > Whether this is easier or harder to understand than alternative representations is a (super interesting), but orthogonal question.
> > However, to answer more directly, the concepts exist and are functionally important (the importance measure makes those concepts in the top-3 among all the classes we investigated), and we haven’t found a class without an Elsewhere concept.
> >
> > Interestingly, we believe it also illustrates a significant limitation of “concept maps”. Concept maps assume concepts fire where they detect features. Elsewhere concepts violate this assumption: they activate off-object yet depend causally on the object's presence.
> >
> > Regarding steering, we do not claim Elsewhere concepts are better for steering than prior approaches. We claim they are a distinct phenomenon that any complete understanding of DINO must account for.
> >
> > ---
> >
> > Thank you once again for your time and engagement with this work. We look forward to incorporating your feedback and believe the manuscript will be improved as a result.

---

### Meta-Review · Area_Chair_LoDW · 2025-12-29

**Summary:**

This paper operationalises the Linear Representation Hypothesis on DINOv2 using sparse autoencoders to extract a large concept dictionary. It analyses how downstream tasks recruit these concepts, identifying distinct functional behaviours such as Elsewhere concepts for classification and border detectors for segmentation. Besides, the work observes deviations from the standard linear hypothesis, motivating the proposal of the Minkowski Representation Hypothesis (MRH), which posits that token embeddings lie in sums of convex polytopes. AC considers this a significant contribution to the field of interpretability. The empirical analysis of task-specific concept recruitment provides concrete insights, particularly the discovery of object negation mechanisms. While the MRH is introduced as a working hypothesis, the authors provide theoretical justification rooted in multi-head attention mechanics and support it with preliminary quantitative evidence regarding geodesic paths and archetypal reconstruction. Most reviewers are positive to its contributions, therefore, I would give a recommendation of accept.

**Reviewer Concerns:**

The authors have answered the questions raised during the discussion, particularly regarding the connection between the experiments and the new theory. They satisfied Reviewer `nELv` by adding hard numbers, such as the geodesic analysis and archetypal analysis. While Reviewer `PH2n` worried that testing only three tasks was not enough to prove a general theory, the authors argued these were the most relevant uses for this model. The authors clarified that their theory comes from the basic math of attention heads. Reviewer `vpmQ`’s technical corrections regarding the charts and native representation constraints were accepted, and the authors clarified for Reviewer `F23h` that while they cannot yet fully explain why the model chooses specific concepts for specific tasks, the observation itself is a solid starting point for future research.

**Reviewer Scores:**

Reviewer `F23h` and Reviewer `vpmQ` are strong supporters with high scores of 8. Reviewer `nELv` gave a positive score of 6 and will likely maintain or improve this rating, as the authors provided the specific math proofs requested to back up the theory and clarified the logical flow of the paper. Reviewer `PH2n` is leaning negative with a score of 4, mainly concerned that the empirical evidence is too limited to support the new hypothesis.

---

### Decision · Program_Chairs · 2026-01-26

Accept (Poster)